# Using citizen science to determine if songbird nesting parameters fluctuate in synchrony

**Sara E. Harrod**[ID]*[⊗], **Virginie Rolland**[⊗]

Department of Biological Sciences, Arkansas State University, Jonesboro, Arkansas, United States of America

⊗ These authors contributed equally to this work.
¤ Current address: John Martin Reservoir, US Army Corps of Engineers, Hasty, Colorado, United States of America
* harrodsara@yahoo.com

**Data Availability Statement:** The data underlying the results presented in the study are available from The Cornell Lab of Ornithology's NestWatch Program (https://nestwatch.org/).

## Abstract

As global temperatures continue to rise, population or spatial synchrony (i.e., the degree of synchronization in the fluctuation of demographic parameters) can have important implications for inter- and intraspecific interactions among wildlife populations. Climatic fluctuations are common drivers of spatial synchrony, and depending on the degree of synchronization and the parameters impacted, synchrony can increase extinction probabilities. Although citizen science is an inexpensive method to collect long-term data over large spatial scales to study effects of climate changes on wildlife, few studies have used citizen science data to determine if this synchrony is occurring across populations and species. We used 21 years of citizen science nesting data collected on Eastern Bluebirds (*Sialia sialis*) and Carolina Chickadees (*Poecile carolinensis*), two widespread North American species with similar life histories and abundant data, to assess the degree of synchrony between and within their populations in the southeastern United States. We found little evidence of synchronous fluctuations in the nesting parameters of hatching success, hatchability, and fledging success between and within species, nor did we observe consistent patterns towards increased or decreased synchrony. Estimates of nesting parameters were high ($\geq 0.83$) and showed little variability (relative variance $\leq 0.17$), supporting the hypothesis that parameters that strongly contribute to population growth rates (i.e., typically fecundity in short-lived species) show little interannual variability. The low variability and lack of synchrony suggest that these populations of study species may be resilient to climate change. However, we were unable to test for synchronous fluctuations in other species and populations, or in the survival parameter, due to large gaps in data. This highlights the need for citizen science projects to continue increasing public participation for species and regions that lack data.

## Introduction

Studies of how populations fluctuate require long-term data ($\geq 10$ years) [1], but spatially large-scale studies are also important because many species are found across various habitats and climatic conditions, and populations may respond to environmental factors such as climate change differently [2–4]. Studies focusing on sub-populations and small regions are

**Funding:** The authors received no specific funding for this work.

**Competing interests:** The authors have declared that no competing interests exist.

common, but inadequate for understanding population dynamics across a species' range. For example, from 1966–2019, House Finch (*Haemorhous mexicanus*) populations in the eastern United States increased, whereas many populations in the southwestern United States declined [5]. In fact, for most widely distributed species, responses to environmental changes (including climate change) likely vary among populations [4] and regions, though common environmental changes experienced by nearby populations can also induce synchronous fluctuations in their demographic parameters [6].

In contrast to seasonal or phenological synchrony (i.e., synchrony in the timing of life history events such as breeding), population or spatial synchrony measures the degree of covariation in the fluctuations of demographic parameters. Fluctuations may occur synchronously (i.e., in positive or negative correlations) or asynchronously (i.e., in uncorrelated ways; Fig 1). Previous studies have documented synchronous fluctuations in abundance for wide-ranging species, including avian species [6, 7]. The driving forces behind these synchronous fluctuations are dispersal and environmental effects, including the Moran effect [8], which proposes that fluctuations among populations with the same density-dependent structure are synchronized by strong density-independent factors (typically weather-related) [9, 10]. Although the Moran effect was developed to explain fluctuations occurring simultaneously among geographically distinct populations of a single species, it can also be applied to closely related species [11], species with similar density-dependent population structures [9, 12], and species with similar life histories [12]. Specifically, nesting parameters (e.g., number of offspring produced per breeding attempt, hatching success) may be synchronized across populations and similar species [12]. This synchrony can be driven by factors such as food availability, extreme climatic events, and competition for food and nesting sites, as was proposed for seabirds by Lahoz-Monfort et al. 2013 [12].

Increased synchrony in nesting parameters could have several impacts on avian populations. Because fecundity, or the number of fledged female offspring produced per breeding female, often contributes strongly to population growth rates among short-lived species such as passerines [13, 14], increased synchrony in fecundity can increase extinction probability and lessen the potential for demographic rescue [6–8]. Consider the following scenario: the nesting parameters (which contribute to fecundity) of two populations of the same passerine species become synchronized, and these parameters contribute strongly to each population's growth rate. If these parameters are negatively affected such that they decline, the population growth rates (and subsequently population sizes) would also decline, and extinction probabilities increase. Furthermore, demographic rescue via immigration into these populations would be less likely to occur, as there are fewer individuals in each population to disperse. As global threats to wildlife (including rising global temperatures and habitat loss) continue to expand, it is important to determine which parameters are synchronized, and whether these synchronizations occur within and/or between species. For these studies, collecting long-term data across large spatial scales is necessary.

Although long-term, large-scale studies are time consuming and logistically difficult to conduct, they are crucial to understanding how environmental factors affect widespread species. Because traditional long-term, large-scale scientific studies (i.e., studies in which data are collected over large geographical scales by trained researchers with specialized education and experience in their field of study) of populations are scarce, an alternative is to use data from citizen science programs (i.e., programs in which individuals from any background collect data over varying geographical scales for use by professional scientists), which allow researchers to gather long-term data from large spatial expanses [15]. Several studies have used such data to assess species' distributions and abundance [16–18], reproductive success [19, 20], the spread of invasive species [21], global variability in auditory signals [22], and phenological

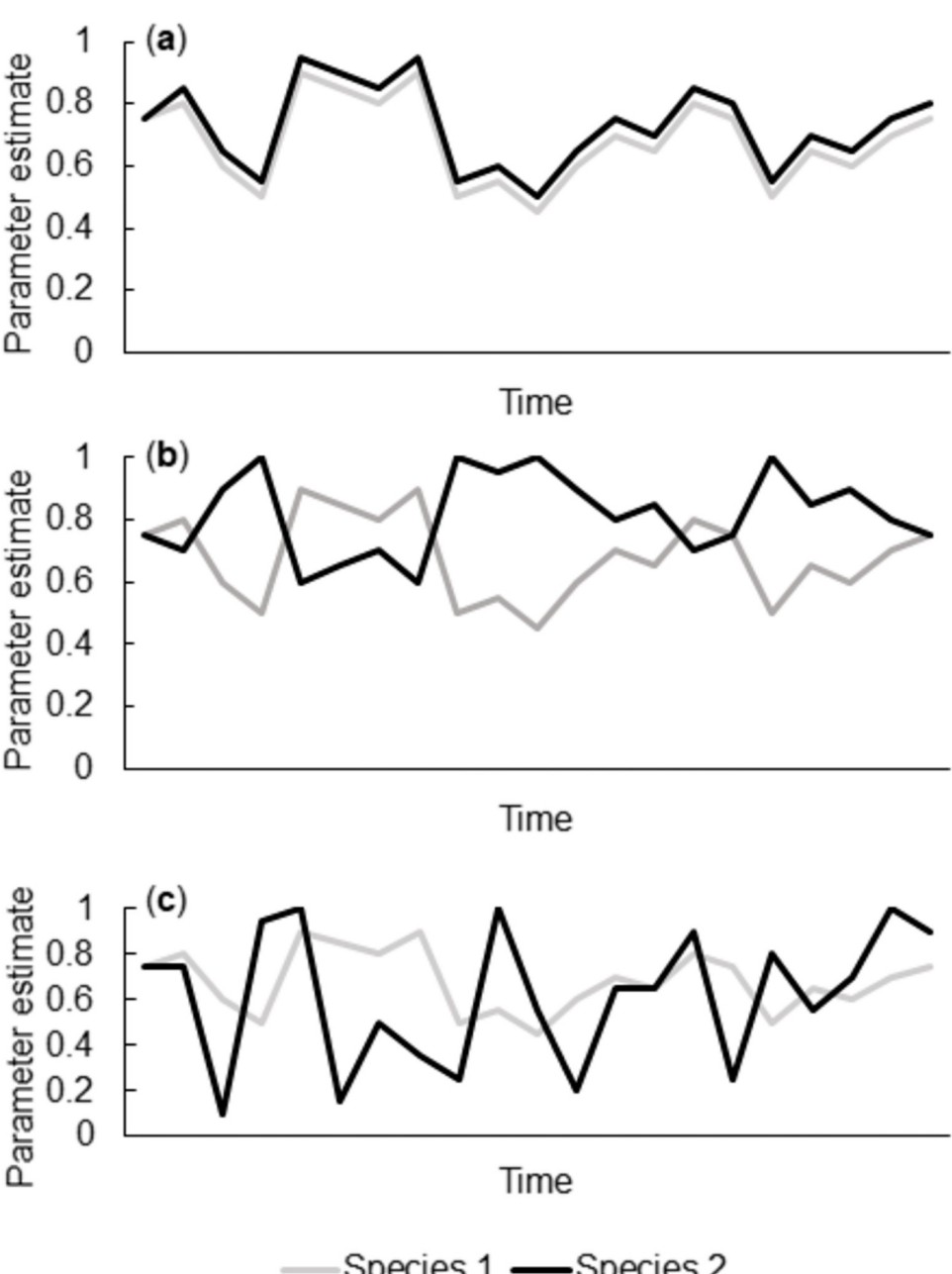

**Fig 1.** A fictitious scenario in which two species exhibit (a) positive synchronous fluctuations (r = 1.00), (b) negative synchronous fluctuations (r = -1.00), and (c) asynchronous fluctuations (r = 0.00) in a given demographic parameter.

shifts [23]. To our knowledge, few studies have used citizen science data to determine if demographic parameters (e.g., survival, nesting success) fluctuate in synchrony across populations and species (but see [24, 25]).

To examine how nesting parameters between and within species respond to climate change and other changing environmental factors, we used 21 years of citizen science nesting data from The Cornell Lab of Ornithology's NestWatch Program [26] for two southeastern US cavity-nesting birds: Eastern Bluebirds (*Sialia sialis*, hereafter 'bluebirds') and Carolina Chickadees (*Poecile carolinensis*, hereafter 'chickadees'). Our objectives were to (1) estimate annual

nesting parameters of bluebird and chickadee populations within and between regions, (2) assess the level of population synchrony in parameters within and between species, and (3) discuss some benefits and constraints of using citizen science data for assessing long-term and large-scale ecological questions. Sæther and Bakke (2000) [14] found that nesting parameters contribute more to population growth rate in short-lived, highly productive avian species than in longer-lived species, and that those parameters with a higher contribution had low temporal variability. Therefore, we predicted that because our study species are short-lived (compared to other avian species) and have large clutches, their nesting parameters would have low temporal variability and thus little synchrony. Understanding whether nesting parameters are synchronized (and to what degree) can provide information about the resiliency of these populations to environmental variability.

## Methods

We obtained NestWatch data for 1998–2018 for bluebirds and chickadees. Though they belong to different families (Turdidae [bluebirds] and Paridae [chickadees]), these species share common life histories. Individuals are typically short-lived, with an average life expectancy of one to two years, their diets consist primarily of insects, and they occur in open woodlands, suburban parks, and residential yards across the eastern United States [27, 28]. These species are secondary-cavity nesters that breed in natural cavities or nest boxes. However, bluebirds lay four to five eggs per clutch and have up to three successful broods per season [28], whereas chickadees typically lay five to six eggs per clutch and have one successful clutch per year [27].

In addition to similar life histories, we chose these species because they are widely distributed across the eastern United States, allowing us to assess parameter synchrony across a large spatial scale and draw more informative conclusions at the species level. These species also readily use nest boxes, so we assumed sample sizes would be large because observers would not have to search for nests. Data were collected in the US Environmental Protection Agency (EPA) Eastern Temperate Forest Level I ecoregion [29], spanning 75.8–97.3˚W longitude, or eastern North Carolina to central Oklahoma (Fig 2). We restricted our study to nests along ~36˚ N latitude to avoid a latitudinal effect of temperature. Furthermore, at this latitude, annual precipitation is comparable (i.e., within 100–152 cm; [30]) and populations are generally non-migratory [27, 28].

To assess nesting parameter synchrony within and across species, we classified nests, based on their locations, between two regions as defined by the Migratory Bird Joint Ventures (MBJV) [31]: Atlantic Coast and Central Hardwoods (Fig 2). We used MBJV regions instead of EPA ecoregions for two reasons. First, unlike EPA Level II ecoregions, MBJV regions were defined not just by using biotic and abiotic data, but also by similarity of avian communities and resource management issues [33]. Second, EPA Level III and IV ecoregions are smaller [29], which would have resulted in many groups with too few nests (average of <10 per year) for analysis. Although we initially considered five additional MBJV regions for our analyses, we could not include them because they contained multiple years with an average of < 10 nests per year. The data underlying this article were provided by The Cornell Lab of Ornithology's NestWatch Program by permission and collected using the Breeding Biology Research and Monitoring Database Field Protocol. Data will be shared on request to the corresponding author with permission of The Cornell Lab of Ornithology.

### Estimating nesting parameters

After removing nest records (up to 721 per parameter) that contained errors (e.g., user reported zero eggs hatched but five chicks fledged), we calculated hatching success (i.e.,

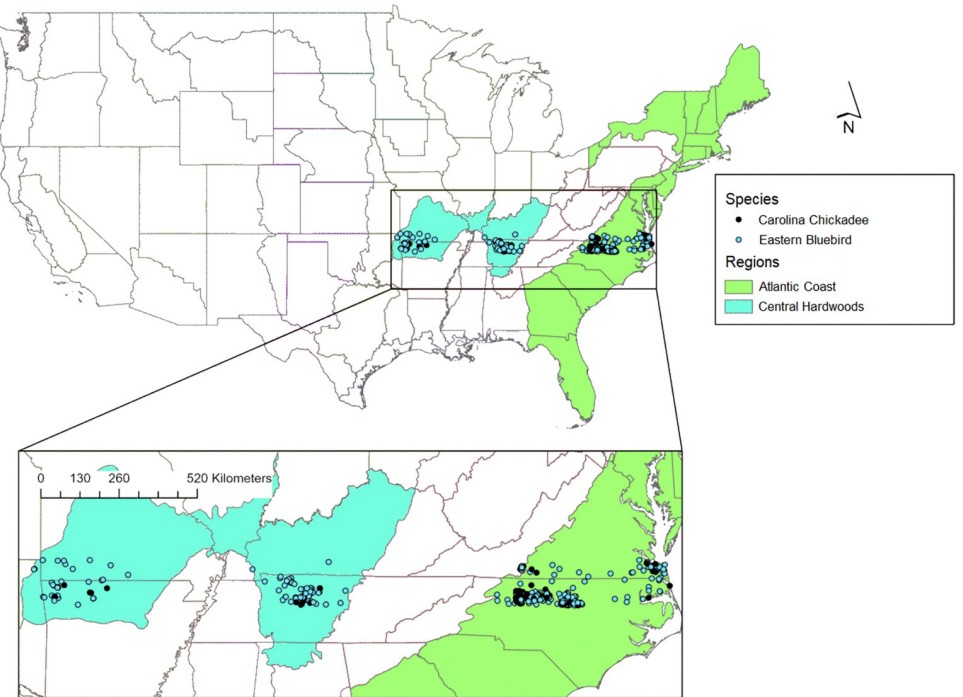

**Fig 2. Nest locations per region [31].** Nesting data for Eastern bluebirds and Carolina chickadees were collected by NestWatch participants, 1998–2018. Each point represents the site of at least one nesting attempt. The Migratory Bird Joint Ventures region shapefile was provided by the U.S. Fish and Wildlife Service [32].

probability ≥ one egg in a clutch hatched), hatchability (i.e., proportion of eggs hatched in a nest in which ≥ one egg hatched), and fledging success (i.e., probability ≥ one chick fledged from a nest in which ≥ one egg hatched). The only nesting parameter that we did not consider was clutch size, because this parameter typically varies more between the first and last clutch of a season for bluebirds than it does between years [34]. We estimated parameters for each species within a region (e.g., AC bluebirds) and for each species across both regions.

We estimated annual parameters using "year" as an explanatory factor variable. For hatching and fledging success, the response variable was binary (1 indicated at least one egg hatched/chick fledged, 0 indicated that no eggs hatched/chicks fledged), whereas hatchability was treated as proportion data with a two-vector response variable—one vector for the number of hatched eggs per nest and the other for the number of unhatched eggs per nest. Ideally, hatching and fledging success should be estimated with logistic-exposure models [35]. Although we obtained daily nest visit data, the intervals between nests visits were wider (ranging from every day to over two weeks) than those recommended for logistic-exposure models (i.e., 3–5 days; [35]). We instead built generalized linear mixed models (GLMMs) with a binomial error distribution using the lme4 package [36] in program R v.3.5.1 [37]. With this approach, estimates may be biased, but the bias would be consistent across species and regions and therefore would not affect further analyses that used these estimates. To account for pseudoreplication among nest location, we included the nest location as a random variable (S1 Table). All estimates are reported ± 1 SE.

Before assessing parameter synchrony, we checked for temporal trends in parameters using linear models of annual estimates from the GLMMs for each parameter against years. Because no parameters at this level showed a significant (α = 0.05) temporal trend (β ≤ 0.006, P ≥ 0.07), detrending was not necessary. Finally, we tested whether parameters remained relatively

stable over time (i.e., canalized against temporal variability) as the environmental canalization hypothesis predicts that parameters that strongly contribute to population growth rate show little interannual variability [38]. For short-lived species such as bluebirds and chickadees, these are often nesting parameters [13, 14]. We therefore measured the relative variance (RV) of parameters at each level, as follows:

$$RV = \frac{\sigma 2}{\mu * (\text{maximum value} - \mu)}$$

where $\sigma^2$, $\mu$, and maximum value are the variance, mean, and highest annual value for a parameter, respectively [39]. We also measured the coefficient of variation (CV) of our parameters as follows:

$$CV = \frac{\sigma}{\mu}$$

where $\sigma$ and $\mu$ are the standard deviation and mean value for a parameters, respectively [39].

For both measures, values near zero indicate low variability, whereas values closer to one (or $> 1$ for CV) indicate high variability.

### Assessing synchrony in nesting parameter fluctuations

We conducted a sliding-window correlation [40] with Kendall tau rank-order correlation coefficients ($\tau$) to test for synchrony in nesting parameters using annual estimates from the GLMMs at the species (one species across the CH and AC regions) and region (bluebirds and chickadees within one region) levels. Sliding-window correlation analyses are conducted for partially overlapping successive years within a window to detect correlations within a portion of the sample, which otherwise could be hidden by the overall correlation [40]. For example, to determine if hatching success covaries between two populations for the period of 1999–2005, a five-year window can be used to test for correlations over the overlapping sub-periods 1999–2003, 2000–2004, and 2001–2005. Individual correlation estimates from each window were evaluated at $\alpha = 0.05$.

We used five- and seven-year windows incremented by one year, following previous synchrony studies [41, 42]. A five-year window is the smallest meaningful window size in such analyses, and Durant et al. (2004) found similar patterns in five-, seven- and nine-year windows. We also ran correlation analyses over the entire period for each level. As correlation coefficients cannot be calculated with missing values, we substituted the overall average for years with missing data, with only CH bluebirds and chickadees having one year with no data. We conducted these correlation tests at $\alpha = 0.05$ at the species and region levels. Because [43] recommended against a Bonferroni correction, we instead looked at patterns of biological relevance.

Finally, we assessed whether trends in synchrony (i.e., our sliding-window correlation coefficients) were significant by using linear models of $\tau$ against time (i.e., sliding windows). All estimates are reported $\pm 1$ SE, and trends were considered significant at $\alpha = 0.05$.

## Results

We obtained data from 4685 bluebird and chickadee nests across the AC and CH. Within the AC region, 1726 bluebird nests and 412 chickadee nests were recorded. Likewise, the CH region contained more bluebird nests (2116) than chickadee nests (431).

### Nesting parameters

GLMMs indicated that hatching success, hatchability, and fledging success fluctuated interannually, but average estimates remained high ($\geq 0.83$; Table 1). At the species level, hatching

**Table 1. Mean (± SE), relative variance (RV), coefficient of variation (CV), sample size (*n*), and the minimum and maximum number of nests per year (Range) for hatching success, hatchability, and fledging success of Eastern Bluebirds and Carolina Chickadees in the Atlantic Coast (AC) and Central Hardwoods (CH) regions, 1998–2018.** Means are averages of annual estimates obtained from the generalized mixed models (GLMM). Standard errors were calculated using these mean values. RV and CV measure variability in GLMM-produced annual estimates of our nesting parameters.

| Level | Hatching success | | | | | Hatchability | | | | | Fledging success | | | | |
|---|---|---|---|---|---|---|---|---|---|---|---|---|---|---|---|
| | Estimate (± SE) | RV | CV | *n* | Range | Estimate (± SE) | RV | CV | *n* | Range | Estimate (± SE) | RV | CV | *n* | Range |
| (a) Species | | | | | | | | | | | | | | | |
| Bluebird | 0.89 ± 0.001 | 0.02 | 0.04 | 3295 | 55–336 | 0.91 ± 0.0003 | 0.003 | 0.02 | 2781 | 52–283 | 0.94 ± 0.001 | 0.02 | 0.04 | 2566 | 52–263 |
| Chickadee | 0.87 ± 0.003 | 0.04 | 0.08 | 660 | 15–49 | 0.91 ± 0.002 | 0.03 | 0.05 | 559 | 14–45 | 0.88 ± 0.005 | 0.12 | 0.13 | 546 | 15–44 |
| (b) Species-region | | | | | | | | | | | | | | | |
| AC bluebird | 0.89 ± 0.002 | 0.04 | 0.07 | 1396 | 12–154 | 0.92 ± 0.001 | 0.01 | 0.03 | 1221 | 12–138 | 0.94 ± 0.001 | 0.03 | 0.05 | 1200 | 12–133 |
| AC chickadee | 0.87 ± 0.008 | 0.14 | 0.15 | 288 | 2–28 | 0.91 ± 0.004 | 0.04 | 0.07 | 253 | 1–25 | 0.90 ± 0.008 | 0.17 | 0.14 | 248 | 1–24 |
| CH bluebird | 0.90 ± 0.001 | 0.15 | 0.04 | 1899 | 13–237* | 0.89 ± 0.001 | 0.01 | 0.05 | 1560 | 13–206* | 0.92 ± 0.001 | 0.04 | 0.06 | 1366 | 13–205* |
| CH chickadee | 0.83 ± 0.005 | 0.08 | 0.12 | 372 | 14–26* | 0.93 ± 0.003 | 0.03 | 0.03 | 306 | 9–24* | 0.87 ± 0.007 | 0.12 | 0.14 | 298 | 9–24* |

*The year with missing data was not included in the range

success was slightly higher for bluebirds than chickadees (Table 1A). At the species-region level, hatching success was highest for AC and CH bluebirds (Table 1B). RV estimates for hatching success remained low at all levels but was highest for AC chickadees and CH bluebirds (Table 1B). CV estimates were highest for AC and CH chickadees (Table 1B).

Hatchability at the species level was nearly identical between chickadees and bluebirds (Table 1A). For our combinations of species and regions, hatchability was highest for CH chickadees and lowest for CH bluebirds (Table 1B). As with our hatchability estimates, the RV and CV remained low across all levels, with the highest variability among AC chickadees (Table 1B).

Species-level fledging success was higher for bluebirds than chickadees (Table 1A). The RV and CV estimate for bluebird fledging success was higher than any other parameter at the species level (Table 1A). Within the species-regions combinations, fledging success was highest for AC bluebirds and lowest for CH chickadees (Table 1B). RV and CV values were highest for AC and CH chickadees (Table 1B).

## Synchrony in nesting parameters

For hatching success, none of our overall correlation coefficients were significant (τ ≤ 0.26, $P \geq 0.11$). However, we found two periods for which hatching success was significantly correlated: between CH bluebirds and chickadees in 1998–2002 (τ = 0.95, $P = 0.04$) with the five-year window (Fig 3D), and between AC and CH bluebirds in 2006–2012 (τ = -0.75, $P = 0.03$) with the seven-year window (Fig 3A). We also found significant increasing trends in synchrony of hatching success between AC bluebirds and chickadees (0.03 ± 0.01, $P = 0.003$; Fig 3C), and between CH bluebirds and chickadees (0.04 ± 0.13, $P = 0.009$; Fig 3D) with the seven-year window.

As with hatching success, no overall correlation coefficients were significant for hatchability (τ ≤ 0.22, $P \geq 0.18$), and no significant period-specific correlations were found for this parameter (τ ≤ 0.89, $P \geq 0.06$). However, with five- and seven-year windows we found significant increasing trends in synchrony for AC and CH bluebirds (five-year window: 0.07 ± 0.014, $P \leq 0.001$; seven-year window: 0.06 ± 0.008, $P < 0.005$; Fig 3A) and AC and CH chickadees (0.07 ± 0.01, $P < 0.005$; Fig 3B), and a significant decreasing trend in synchrony for CH bluebirds and chickadees (-0.07 ± 0.01 and -0.06 ± 0.01, $P < 0.005$; Fig 3D). AC chickadee and bluebird hatchability showed few strong correlations, with no discernible pattern or significant trends in synchrony (-0.01 ± 0.02, $P \geq 0.42$; Fig 3C).

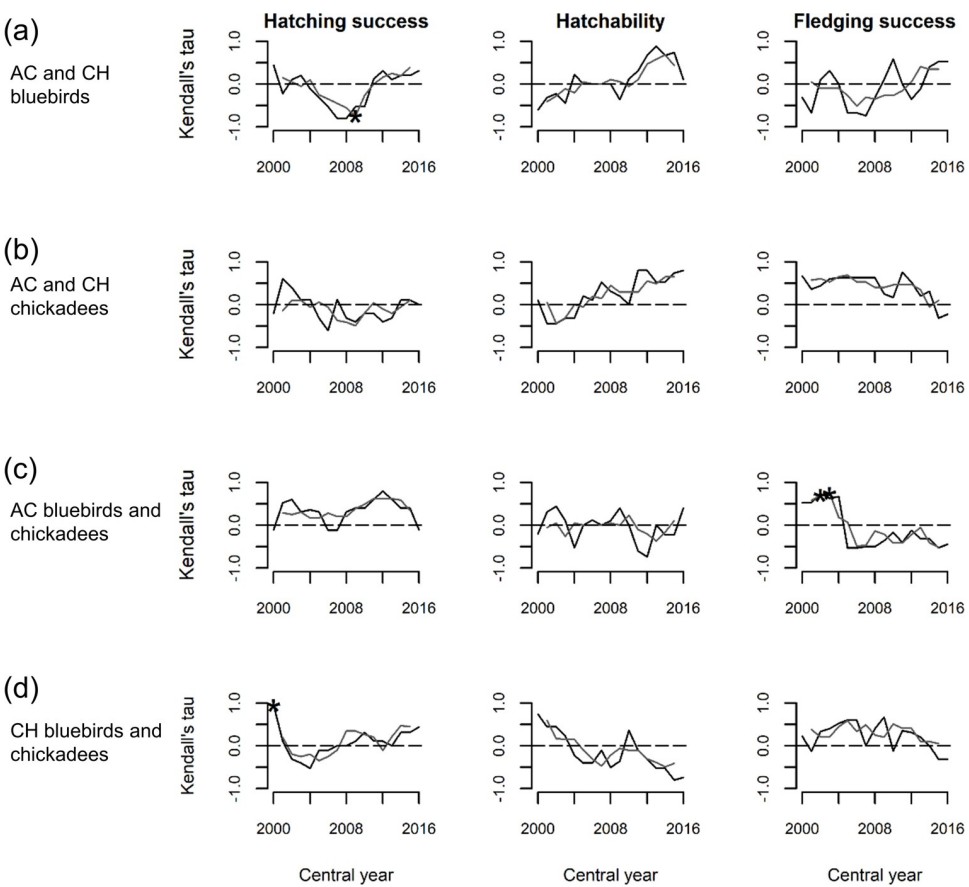

**Fig 3. Five-(black) and seven-year (grey) sliding window correlations of three nesting parameters.** Correlations for hatching success, hatchability, and fledging success were calculated between (a) Atlantic Coast and Central Hardwoods bluebirds, (b) Atlantic Coast and Central Hardwoods chickadees, (c) Atlantic Coast bluebirds and chickadees, and (d) Central Hardwoods bluebirds and chickadees. The central year of each window is displayed on the x axis; for example, the year 2000 for the five-year window represents the period 1998–2002. Significant periods of correlation are indicated by asterisk.

For fledging success, the overall correlation coefficient was significant between AC and CH chickadees ($\tau$ = 0.46, $P$ = 0.01; Fig 3B). We also found two significant correlations for fledging success between AC bluebirds and chickadees in 1999–2005 ($\tau$ = 0.70, $P$ = 0.05) and 2000–20006 ($\tau$ = -0.75, $P$ = 0.04) with the seven-year window (Fig 3C). With the five-year window we found a significant increasing trend in synchrony for fledgling success between AC and CH bluebirds (0.05 ± 0.02, $P$ = 0.04; Fig 3A). Using both windows, we found significant decreasing trends in synchrony for fledging success between AC and CH chickadees (five-year window: -0.04 ± 0.01, $P$ = 0.01; seven-year window: 0.07 ± 0.01, $P$ < 0.005; Fig 3B) and between AC bluebirds and chickadees (-0.07 ± 0.02, $P$ < 0.005; Fig 3C). No significant synchronous trends were found for CH chickadees and bluebirds (five-year window: -0.01 ± 0.01, $P$ = 0.14; seven-year window: -0.03 ± 0.01, $P$ = 0.08; Fig 3D).

## Discussion

As worldwide threats such as climate change and habitat loss continue to impact wildlife populations, studies are needed to assess how wildlife demographic parameters are responding. To make robust predictions, long-term and large-scale data are needed, since populations may

respond differently [4], fluctuate in synchrony [12], or be canalized against environmental variability [38]. These data can reveal if population synchrony is occurring within and across species, and the degree of inter- and intraspecific synchrony can allow researchers to predict extinction risk in separate but synchronized populations. In this study, using 21 years of citizen science data on two species across the eastern United States, we determined if fluctuations in nesting parameters were spatially synchronized between and within species. Synchronized parameters across populations and species could have negative consequences such as increased risk of extinction and/or lower chances of demographic rescue [6–8], particularly if the parameters strongly contribute to population growth rate and are not canalized (i.e., they fluctuate strongly). For example, if a given demographic parameter is synchronized between two interspecific populations, a decline in the parameter of one species could inform researchers that the other species is likely experiencing a decline in this parameter as well. Overall, we found no consistent patterns of synchrony across nesting parameters between species or regions. Hatching success exhibited the fewest significant synchronous trends. Hatchability had no synchronous windows at any level, but exhibited significant trends toward increasing and decreasing synchrony at for both species and one region. Chickadees exhibited significant synchrony in fledging success across regions, but this synchrony decreased over time.

RV and CV estimates were low ($\leq 0.17$) for each parameter at each level, suggesting that, as predicted, these parameters might be canalized against temporal variability. Additionally, only one parameter (bluebird hatchability at the species level) exhibited a significant temporal trend, further supporting the environmental canalization hypothesis that parameters which strongly contribute to a population's growth rate show little interannual variability [38, but see 44 for effects of increasing temperatures on hatching success]. Cavity-nesting birds tend to have higher nesting success than non-cavity nesters [45]. In particular, predator guards such as stovepipe baffles and entrance hole extenders, improve nest survival [19] and could have also canalized our nesting parameters, particularly hatching and fledging success, and could explain the relatively low RV and CV values. Because predator guard data were unavailable, we were unable to determine which nests were protected.

The same synchrony patterns were present with the five- and seven-year windows, but more significant correlation coefficients were detected with the seven-year window. Generally, as expected for short-lived, highly productive passerines who nesting parameters exhibit little temporal variability [14], we rarely detected significant periods of synchronization (but see Fig 3A, 3C and 3D) or consistent patterns toward positive or negative synchrony among and within bluebird and chickadee populations. Although synchrony for fledging success between AC and CH chickadees has continuously declined over the past 20 years, this is the only parameter for which we found a significant overall correlation in three out of four species and region levels. Therefore, we cannot conclude that synchrony is occurring between and within bluebird and chickadee populations in the AC and CH regions; if synchrony is occurring, it may be too weak to detect.

The lack of significant correlations could reflect that there is no nesting parameter synchrony and could be due to the low variability among our parameters. Nesting parameters in short-lived species are more canalized than survival [14], and RV and CV estimates for each parameter were low (though this could be due to predator guards). As survival tends to be less canalized for short-lived species [1], this parameter may be synchronized within and between species. However, collecting survival data requires bird banding (for which citizen scientists generally do not hold a permit) and we are not aware of a citizen science program that collects capture-recapture data. Additionally, resighting banded birds, and recovery rates are often low [46]. Nonetheless, we detected eight significant trends toward increased (AC bluebird and chickadee hatching success, CH bluebird and chickadee hatching success, AC and CH bluebird

hatchability and fledging success, and AC and CH chickadee hatchability) and decreased (CH bluebird and chickadee hatchability, AC and CH chickadee fledging success, and AC bluebird and chickadee fledging success) synchrony in our nesting parameters. Furthermore, although parameters had low variability over the study period, the variable annual number of nests (1–283) may have led to imprecise estimates. The lack of consistent synchrony suggests that despite facing similar climatic conditions, these fluctuations of nesting parameters between bluebirds and chickadees are independent. This bodes well for these populations since their risk of extinction is not heightened by increased population synchrony, except for two slight increases in hatching success synchrony. Within species, however, the decrease in fledging success synchrony between AC and CH chickadees may be countered by the increase in hatchability synchrony between AC and CH chickadees. Therefore, if fledging success is high in one population, it may be a result of fewer chicks hatching, and though this would potentially sustain the population with high fledging success, it may not be enough to repopulate the population that crashed.

To make this study more informative at the avian community level, we could have included other secondary cavity-nesting species with similar traits to bluebirds and chickadees. For example, Tufted Titmice (*Baeolophus biocolor)* and Carolina Wrens (*Thryothorus ludovicianus)* also have a widespread and similar distribution, tolerate human activity, and frequently use nest boxes and other artificial sites (e.g., mailboxes and flowerpots) [47, 48]. Given the proximity of these passerines to humans in one of the fastest growing US regions [49, 50], we were surprised by the large data gaps and small sample sizes across regions. With these limitations, we could focus on only two species nesting in two of the seven regions considered.

Although citizen science projects have grown over the past decade [51–53], and their data are invaluable for researchers, gaps remain for several species and regions. We suggest citizen science projects identify data-deficient regions and focus on bolstering participation. Additionally, untranscribed historic data could help fill gaps. Websites such as Zooniverse [54] allow participants to transcribe recent and historical scientific data. Nest Quest GO!, launched by the NestWatch Program, allows individuals to transcribe records from the North American Nest Record Card Program, which ran from the 1960s to early 2000s. This program contains records for >17 avian species, including Prothonotary Warblers (*Protonotaria citrea*), Mountain Bluebirds (*Sialia currucoides*), American Kestrels (*Falco sparverius*), Carolina Chickadees, American Woodcocks (*Scolopax minor*), and Tufted Titmice. Significant progress has been made, with transcription completed for several species. However, new initiatives alone are not enough to bolster public involvement. We encourage project leaders and scientists to increase public awareness about citizen science projects and transcription programs so that new and historic data can be collected. In particular, common barriers to data collection should be addressed appropriately. For example, a lack of data for a given species due to the species rarity could be remedied by not only increasing awareness of nest monitoring programs, but by encouraging citizens to provide appropriate nesting habitat for this species. With more data, it may be possible to re-test for synchrony in nesting parameters among these species and regions and determine if our results are due to absence of synchrony or lack of data. Such studies should also be conducted among other geographic regions and species, particularly those most affected by anthropogenic disturbances.

## Supporting information

**S1 Table.** Summary of random-effect model selection for (a) hatching success, (b) hatchability, and (c) fledging success of Eastern bluebirds and Carolina chickadees at the species and species-region levels. No fixed terms were included. Though the model with nest location was

not always the most parsimonious, we included this random variable in all our models for consistency across analyses. Models are presented with corresponding k (number of parameters), AIC (Akaike's Information Criterion), ΔAIC (difference in AIC between given model and lowest AIC model), and w*i* (model weight) values.
(DOCX)

## Acknowledgments

We thank the volunteers who collected the nesting data for this study through The Cornell Laboratory of Ornithology's Nest Box Network, The Birdhouse Network, NestWatch, the Smithsonian Migratory Bird Research Center's Neighborhood NestWatch, and various state nest monitoring projects that have contributed their historic data. We also thank T.D. Marsico, V.S. Kulkarni, and T.J. Boves for their helpful comments on earlier drafts of this manuscript, and V. Kearny for creating Fig 2.

## Author Contributions

**Conceptualization:** Virginie Rolland.

**Data curation:** Sara E. Harrod.

**Formal analysis:** Sara E. Harrod.

**Investigation:** Sara E. Harrod.

**Project administration:** Virginie Rolland.

**Supervision:** Virginie Rolland.

**Writing – original draft:** Sara E. Harrod.

**Writing – review & editing:** Virginie Rolland.

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
