## [Decision Letter · Decision Letter 0]

31 Mar 2022

PONE-D-22-01496Using citizen science to determine if songbird breeding parameters fluctuate in synchronyPLOS ONE

Dear Dr. Harrod,

Thank you for submitting your manuscript to PLOS ONE. After careful consideration, we feel that it has potential merit but does not meet PLOS ONE’s publication criteria as it currently stands. Therefore, we invite you to submit a revised version of the manuscript that addresses the points raised during the review process. The revision required is likely to be fairly extensive. All three reviewers and myself agree that investigating spatial synchrony on demographic parameters is a worthy goal and would be of wide interest.  However, the reviewers each identified a number of shortcomings that will need to be addressed.  These involve making the conceptual background for the study richer and deeper, not focusing so much on methods and why trends weren't significant, and addressing issues of study and analytic design; in addition, there are a number of methodological clarifications or modifications that need to be made.  The reviewers made many useful suggestions, both more general and specific, that you will want to address in your revision.

We look forward to receiving your revised manuscript.

Sincerely,

Charles R. Brown

Academic Editor

PLOS ONE

Journal Requirements:

2. We note that Figure 2 in your submission contain map images which may be copyrighted. All PLOS content is published under the Creative Commons Attribution License (CC BY 4.0), which means that the manuscript, images, and Supporting Information files will be freely available online, and any third party is permitted to access, download, copy, distribute, and use these materials in any way, even commercially, with proper attribution. For these reasons, we cannot publish previously copyrighted maps or satellite images created using proprietary data, such as Google software (Google Maps, Street View, and Earth). For more information, see our copyright guidelines: http://journals.plos.org/plosone/s/licenses-and-copyright.

Reviewers' comments:

Reviewer's Responses to Questions

**Comments to the Author**

1. Is the manuscript technically sound, and do the data support the conclusions?

Reviewer #1: Yes

Reviewer #2: Yes

Reviewer #3: Partly

2. Has the statistical analysis been performed appropriately and rigorously? 

Reviewer #1: Yes

Reviewer #2: Yes

Reviewer #3: No

3. Have the authors made all data underlying the findings in their manuscript fully available?

Reviewer #1: Yes

Reviewer #2: Yes

Reviewer #3: No

4. Is the manuscript presented in an intelligible fashion and written in standard English?

Reviewer #1: No

Reviewer #2: Yes

Reviewer #3: Yes

5. Review Comments to the Author

Reviewer #1: The authors conducted a new and interesting study on if synchrony of demographic parameters exists within and across populations of two bird species using citizen/community science data. The authors demonstrated strength in describing their methods and clearly stating their results. My main concern with the manuscript is that the authors did not explain why a broad-scale analysis of synchrony should be studied and that they centered their manuscript on using the citizen/community science dataset rather than how synchrony could affect bird populations. The authors need to refocus the manuscript on the importance of synchrony as it pertains to bird conservation and why it is important that we should examine synchrony across multiple populations rather than look at demographic parameters of individual populations. The conclusions of the manuscript are difficult to discern; the majority of the discussion focuses on gaps in the dataset or alternative explanations as to why few significant results were found. The authors need to provide broader context for increased synchrony in hatchability and how that could be a benefit under changing environmental conditions.

Major Revisions

Abstract

Lines 22-25: I would start your abstract introducing what synchrony is and how climate change is affecting it rather than stating the benefits of citizen science. You state that no study has examined synchrony across populations and species but you do not define what synchrony is or why it is important.

Lines 26-27: Please state the name (common and scientific) of the species you analyzed. You also need to state that hatching success, hatchability, and fledging success are the breeding parameters you measured; you use the term breeding parameters in the next sentence but you need to state that this term relates to the dependent variables mentioned above.

Lines 28-31: Your conclusions could use some more support; why would breeding parameters be buffered for temporal variation? Are there any other alternative explanations as to why synchrony was not found between populations? You transition into the factors that could have affected your study rather than focusing on what your conclusions say.

Introduction

Lines 36-37: Your first sentence has a strange transition. You could combine the first two sentences of your introduction to get the point across that long-term and large-scale data are needed to determine how environmental factors and climate change could affect species’ demographic parameters.

Line 41: Any songbird examples of the subpopulation or population level not providing adequate information on species responses to change? While the penguin example is good, it is more relatable to have a songbird example since that is what you are studying. Also, the scientific name needs to be in parentheses.

Lines 45-86: I would restructure these paragraphs so that your synchrony paragraphs (54-86) goes before the citizen/community science paragraph (45-53). While the citizen/community science program is important and puts an emphasis on using these dataset in large-scale studies, the main focus of your study is how synchrony affects demographics of species and populations.

Lines 65-66: State what specific breeding parameters may be synchronized across populations and similar species.

Lines 87-89: Revise sentence to “To examine how breeding parameters between and within two southern cavity-nesting bird species respond to climate change and other changing environmental factors, we used 21 years of community science nesting data from The Cornell Lab of Ornithology’s NestWatch Program.” or something similar.

Discussion

Lines 251-253: You restate your question which is good, but you need to clearly state your conclusions, which I believe begin on line 256.

Line 254: State what these negative consequences could be.

Lines 268-269: From what I understand hatchability synchrony increased significantly over the past 20 years. You then spend the next few paragraphs discussing gaps in data, examining other species, and the benefits of citizen/community science projects. You need to expand on this result that hatchability synchrony has increased for bluebirds, and what the ramifications are in the context of environmental change and/or climate change.

Minor Revisions

Introduction

Line 73: Avoid starting sentences with “because.” You could use the word “since” instead.

Methods

Lines 103-105: Revise sentences so there is similarity between sentence clauses.

Line 125: After stating the ecoregions, put abbreviations in parentheses after. You use these abbreviations in Table 1; they should be used throughout the rest of the manuscript.

Line 163-164. Consider adding an Equation section at the end of the manuscript and having an Equation 1 label here. You also need to state what each of the variables mean in this equation.

Reviewer #2: This is an interesting paper on a relatively understudied topic. While studies of spatial synchrony in population size are becoming common, studies that investigate synchrony in the underlying demographic rates (e.g., reproductive output) remain rare. The authors took an innovative approach, through the use of a citizen science program, to shed light on a fundamental question in this sub-field: is there empirical evidence for such synchrony in demographic rates within and among two ecologically-similar species. They present a compelling case study of two cavity-nesting insectivores in the Southeast US. I think the Introduction and especially the Discussion would benefit from a careful revision to improve flow and to provide deeper context into why such synchronous dynamics are of interest. For example, the authors touch on an interesting hypothesis related to the lack of variability in some reproductive rates, but they fail to develop the idea fully for the uninitiated reader. Similarly, I found the presentation of the analytical details in the Methods lacked clarity in some places, and I suggest alternative approaches in others. The most significant omission, I believe, is the lack of adequate discussion of the role that uncertainty in the reproductive parameters could play in producing “non-significant” correlations. There is variability in the number of nest observations available to estimate these parameters among years, and this variability is not currently reported adequately. While I don’t think a full-scale propagation of uncertainty (e.g., a Bayesian approach) is absolutely necessary in this case, the possibility should be at least more fully discussed. These suggestions and a number of smaller concerns related to unclear language are detailed in the line-by-line comments below.

Abstract

The abstract is well written but lacks any numerical results. I understand there were few significant effects, but perhaps reporting about the generally high success and low variability observed would be useful.

L29: The phrase “buffered against temporal variability” requires more explanation. This may be where you can insert language about low variability.

L32-33: suggest deleting this phrase: “to promote the transcription of historic data.”

L63: suggest changing “occurring simultaneously in intraspecific populations” to “among geographically-separated populations of a single species” or something similar.

L65: The phrase “breeding parameters” seems a little vague throughout the ms. Please define it and / or consider using an alternate wording.

L87: The phrase “Because of continuing threats such as climate change…” does not seem to fit with this sentence and I suggest deleting it. But I do think the motivation for the study needs to be clearly stated further down in the paragraph.

L89: I suggest naming the two species here. (And also in the abstract or at least in the keywords.)

L91-94: You do not state any hypotheses or predictions. I think it would be helpful to state what you expected to find and why it would be important to measure the things you did. These should then be revisited in the Discussion.

Methods

L111: why does the study span such a narrow band of latitude? I’m assuming it was done to limit the variation in temperature? Are there any other major environmental gradients along this east-west transect (e.g., precipitation)?

L112: citation(s) for non-migratory status?

L118: consider replacing “sorted” with “grouped” or some other word.

L119: I don’t think the software that you used for this is relevant here.

L123-124: “Though EPA level…” I think this sentence could be improved. For example, what is meant by “more precise”. Do you mean smaller? What does “too small for analysis” mean exactly?

L125: “We only included data from the Atlantic Coast and Central Hardwoods region…” Then why show all the nests in all the regions on the map? I suggest only showing the nests and region you studied, and explaining how you arrived at that study are in the text.

In general, I think the description of the study area and how it was chosen is a bit disorganized and contains extraneous information. Please try to revise this section to make it more clear and concise.

L131: Please state how many nest records were removed.

L133-134: “probability ≥ one chick fledged” … Please clarify if this was calculated only for nests that had already hatched, or was it for all nests as is typical for “nesting success” measures. That is, was the denominator all nests of the species or only those that reach the nestling stage?

L134: “probability ≥ one chick fledged” … It is well-established that “naïve” estimates of fledging success (and hatching success too) are biased high due to the inability to discover nests after they have failed. Please consider using an “exposure-days” approach (e.g., logistic exposure modeling) or otherwise assure the reader that this source of bias won’t affect your conclusions. For example, what is the visit interval for the Nest Watch program, and was the visit interval similar among the two species?

L134-135: “all ecoregions” … This was confusing to me as I was under the impression that all analyses were restricted to just the two ecoregions. I think I understand now, that the “species level” included all the nests on the map, while the other two analyses were restricted to the AC and CH. Please consider clarifying how you present this in the text and perhaps in the map figure as well.

L136-137: “Due to low annual…” This is repeated information from L125-126.

L140-141: “…by randomly selecting a subset of bluebird nests…” Does this mean that for each year, the same number of bluebird and chickadee nests were included in the analysis? Or just the same total sample size? Please clarify.

L140-141: While I see the value in equaling the sample sizes to ensure compatibility of the datasets, it also discards information and introduces greater uncertainty in the bluebird estimates. I think the authors should consider a bootstrapping-type randomization approach that repeats this process many times to avoid discarding information. At the very least, repeating the random selection and analysis say 5 times and reporting whether or not the conclusions changed would do a lot to alleviate this concern. The primary results could remain as is, but with the addition of a statement that repeating the process did not influence conclusions.

Table 1. Can you report the min and max annual sample size of nests across years in a new column (or columns) here? The only information I see about annual sample size in the methods is a rule to ensure at least 10 nests per year. But then you mention later that there was one year of no data in the Central Hardwoods. Please clarify. I’m guessing Central Hardwoods has at least 10 nests per year except for the one year which had none? Summary statistics would be helpful.

Table 1. Please clarify how these estimates and SE were produced. Are they model predictions or simple averages across years?

L148: What does “at all levels” mean here? Please clarify. Also please clarify the model structure: did you get the estimates by including year as a categorical variable in the model?

L149-150: What is the definition of “nest site” and “nest location” here? Is it the same nest box? Or the same geographic coordinates? Please clarify. How common were “multiple-nesting-attempt” locations in the data set? On average how many attempts per location?

L149-156: I think including nest location as a random-effect is advisable whether or not it performs better based on AIC. Multiple nesting attempts from the same nest box, for example, seems like a clear example of non-independence. Multiple attempts from the same box within the same year are even less likely to be independent. As this only affects one model in your analyses, I think using a random-effect for all and dropping the language about model selection from this paragraph would streamline things and make for a more consistent and appropriate analysis.

L149: please clarify the “family” of GLM you used to assess these models. I am assuming it was Binomial for the binary data. However, was a Gaussian or Binomial model used for the proportion data? The best way to approach the proportion data, in my opinion, is to use a Binomial GLM with the “weights” argument specifying the total number of eggs in each nest. Here is an example from an online posting: https://stackoverflow.com/questions/50078497/weighted-logistic-regression-in-r

L164: Sigma and mu are not defined here.

L164: Why not report the coefficient of variation (sd/mean) for this (instead of, or in addition to, the other measure)? It is similarly standardized and might be easier for the general reader to interpret.

L172-174: “For example, to determine if hatching success…” I recommend modifying and clarifying this example. On first read, it doesn’t seem appropriate to estimate a correlation over only 3 or 4 years of data. But if this is indeed how the technique works, please expand on this a little more to convince the reader it is appropriate. Are the sliding-window estimates then averaged? Or are they plotted to look for time periods of higher correlation? This is an interesting approach and I think other readers would also benefit from a little more detail about how it is evaluated. Please clarify. Why not look for correlation over the entire time series as well, similar to other population spatial synchrony work?

L179: “combinations with …” Please clarify: combinations of what?

Results

L185: Suggest changing “…Central Hardwoods.” to “…Central Hardwoods regions.”

L199: Again, I think most readers could use some guidance in interpreting “relative variance” (e.g., what is a high value?). Consider using CV.

L207-221: I’m not very familiar with the sliding-window approach, but reports of a few short intervals that were significantly correlated across time do not seem very meaningful to me. I think it would be preferable to report measures of overall time series cross-correlation. Then, where none was found, report that some scattered short intervals were correlated and refer readers to the figure instead of presenting each one. It is easy to get confused by reading a report of each significant “window” without getting the overall context that in general there were few meaningful patterns. Perhaps a table would help to organize these results and allow for a more streamlined section.

Discussion

L251: “In this study, using 21 years of…” Often, the first paragraph of the Discussion includes a concise high-level summary of your results. But if you choose not to go that route (which is fine with me), this sentence seems out of place as it simply states what you studied without providing any results.

L255-259: I would define the term “canalized” again here in the Discussion if you think it is important to include. I think many readers would benefit from a reminder of what it is exactly, and potentially some more background about the hypothesis you reference.

In general, I think the Discussion lacks in-depth speculation about the causes and the implications of having synchronous breeding parameters.

L259: Why is the effect potentially spurious? Also, I thought no time series were found to exhibit a linear time trend?

L266: “but see Figs 2a…” This should refer to Fig. 3 instead.

L267-269: Again, I’m confused as I thought no time series were found to have a trend.

L272-273: One other major factor could have led to non-significant correlations: uncertainty in the breeding parameter estimates themselves which were based on variable amounts of data (n = ~ 10 to ? per year). I realize that it would be a considerable computational challenge to fully address parameter uncertainty in this analysis, but it should at least be discussed. If you found clear patterns, that would argue that parameter uncertainty could safely be ignored. The fact that you did not find clear patterns suggests that it may be important to consider in future analyses. You touch on this in L279-280, but I think it deserves more attention.

L279-280: Please reword: “…is enough to wonder if this result is spurious,…”

L298: Missing an “and” before “Carolina Chickadees”

L290-303: This paragraph discusses the need for more citizen science nest monitoring data without tying it back into how it would benefit our understanding of the synchrony of breeding parameters, which is the focus of the next (very short) paragraph. I suggest merging these two into one concise paragraph, and spending more time on the research implications and less on the various programs and species.

One additional topic of discussion to consider: cavity nesting species are known to have higher nesting success (lower predation rates) than non-cavity nesting species. Could this be one reason you found relatively low interannual variability, and might you find different results with non-cavity nesting species?

Reviewer #3: This paper uses data collected from long term nest recording studies to examine synchronous fluctuations in breeding parameters of two cavity-nesting bird species and populations breeding in different areas of the eastern United States. The authors aim to assess the level of synchrony in breeding parameters between two species each of which have two populations, and to review the benefits of using citizen science data for addressing ecological questions. The authors’ analysis led them to conclude that there is not much evidence of synchrony in selected breeding parameters between species and populations, although due to limitations in the data it is not possible to conclude whether this is due to lack of synchrony or lack of appropriate data. The paper is well written in general, though there are major problems with study design and analysis that makes it difficult to draw conclusions from this work.

Examining synchrony among populations and species is a useful way of understanding how populations and species react to changes in the environment or climate. Synchronous changes in breeding parameters by species or populations suggest reliance on similar habitats or climatic conditions making certain groups of species vulnerable to environmental or climatic change. Although the aim of detecting synchrony in species/populations is useful, it is not clear in this study why these particular species/populations were chosen for comparison, and why attempts were made to detect synchrony in the selected breeding parameters. The paper could be improved by explaining more clearly in the Introduction the purpose of the study in terms of the species/populations/breeding parameters chosen and how the findings contribute to the field of ecology or conservation management.

The authors state in the Abstract that “no studies have used citizen science data to determine if demographic parameters fluctuate in synchrony across populations and species”. The issues with this statement are 1) that the definition of “citizen science” is unclear and 2) that many studies have been published that use data routinely collected by amateur ornithologists or reserve wardens to examine synchrony in demographic parameters across populations and species (e.g. Robertson et al. 2015). Also, one of the study aims stated in the Introduction is to “review the benefits and constraints of using citizen science data for assessing long-term and large-scale ecological questions.” I don’t think this has been adequately covered in the paper. The Discussion does include some discussion about the use of citizen science data, but more discussion could be included on 1) the definition of citizen science data; 2) how widespread the use of data collected by amateur scientists is in studies; 3) the advantages/disadvantages of using these data.

This study may also benefit from an examination of causes of synchrony between species, perhaps using weather/climate data. The analysis as it currently stands just looks for synchrony between species/populations at different time scales. Including an analysis of environmental/weather data that may drive potential synchrony would be interesting and would increase the impact of the study.

The Methods section states that GLMMs were used to estimate breeding parameters (lines 147-151), but the discussion of the results is missing in the rest of the paper. The authors could make it clearer why GLMMs were used in the analysis and how the results were used in the study.

Minor comments:

Lines 25-29: The authors do not make it clear why examining synchrony across these species is important. The abstract begins by focusing on the utility of citizen science, but fails to link this premise with the study aim of examining synchrony in demographics two cavity-nesting species.

Lines 25-29: Not clear here why the authors aimed to compare these two species - were the two species chosen simply because data were available or was there an underlying biological hypothesis driving this study? If so, make explicit here.

Line 29: “Breeding parameters could be buffered against temporal variability”; not clear what this phrase means.

Lines 47-48: The authors need to better define "large-scale scientific studies" and "citizen science programs". There may be some overlap between these (e.g. scientific studies may rely on data collected by volunteer amateur ornithologists, so not clear under these definitions whether these should be defined as “large-scale scientific studies” or “citizen science programs”)

Lines 51-53: It's not clear here what the authors mean by "citizen science data". Many studies have made use of data collected by ringers or bird watchers who were not directly involved in the study, but who collected data that were later used. For example, many studies have been published using long-term monitoring, collected by amateurs or reserve wardens and later analysed to examine synchrony among populations or species (see Robertson et al. 2015, Lahoz-Monfort et al. 2017).

Line 59-60: Could be clearer regarding which phenomenon the authors are referring to here.

Line 64: Add reference after “it can also be applied to closely related species”

Line 65: Add reference after “and species with similar life histories”

Line 65-66: Explain here what factors are driving this synchrony and give examples of how breeding parameters fluctuate across populations and species.

Line 68: In Figure 1 legend the authors could explain how data were simulated in this example.

Lines 73-74: Fecundity is defined as number of young produced per female (see Nagy and Holmes 2004, Journal of Avian Biology 35: 487-491) rather than number of fledged female offspring as stated here.

Lines 75-76: Explain how “increased synchrony in fecundity can increase extinction probability

and lessen the potential for demographic rescue”

Line 77: This phrase is confusing, the authors should be clear what they mean here. Intraspecific is defined as “produced, occurring, or existing within a species or between individuals of a single species”, therefore the phrase “intraspecific passerine populations” doesn't make much sense.

Lines 87-94: This last paragraph attempts to explain why this study was carried out, the specific aims of the project, and how the study contributes to ecology and conservation. I think the authors could make this section clearer - the aims are not well explained and it is not clear how they relate to the text above. What is meant by “two spatial scales” and why is it important? Why did the authors choose these breeding parameters? These are questions that should be addressed here.

Line107: “allowing us to assess breeding parameter synchrony across a large spatial scale.” Explain why is this important.

Line 111: This is a large area. You might expect populations closer together to exhibit more synchronous breeding behaviour compared with species further apart in this range. Did the authors test this hypothesis? This appear to be the case according to text below, but is not made clear in the study objectives in the Introduction.

Line 114: Perhaps the authors could colour points to show number of nests available per year in Figure 2.

Line 122: “MBJV ecoregions were defined using avian ecological data”. Describe how this was done and how this method differs from EPA Level II ecoregions.

Line 126: It looks from Figure 2 that there would have been enough data for analysis in the Appalachian mountain region. The authors could explain here why this was not possible, even for a smaller number of years.

Line 131: Report how many nests per area/year/species contained errors.

Line 132: More often “percentage of eggs which hatch in a clutch” is used to define hatching success. Not clear how the authors calculated the probability that at least one egg hatched. If this is a metric calculated from data, why would a probability be needed? Wouldn't proportion/percentage be correct?

Line 133: What is meant by a “successful” nest?

Lines 140-141: This would reduce the size of your dataset unnecessarily. Perhaps accounting for differences in sample sizes in your statistical analysis would have preserved sample size in this case.

Table 1: Define what is meant by “relative variance” in the table heading. Also, some numbers are overlapping in Table 1

Line 148: It isn't clear here what the authors have done. What was the GLMM used for? What response and explanatory variables were included in the model and what was the reason for doing this? If the purpose is to compare breeding parameters between species/area/year these variables should have been included as explanatory variables. It isn't clear what is meant by “estimated annual parameters” given that in lines 131-133 it is stated that these were calculated from the data.

Line 151: How can you compare a linear mixed model in which one of the models included no random effect? Presumably a different model was used - state here what this was.

Lines 155-156: How many years of data were available? State here.

Line 158: Explain what these linear models were and how they were used in this example.

Line 173: This is a short time series over which to determine whether there is a correlation. Did the authors check to see whether there were enough data points to determine whether there was a significant correlation?

Line 175: Explain why these window sizes were used.

Line 184: Report numbers of nests for each species/area/year here.

Line 187: It is not clear where the results of the GLMMs are in the Results section. What was the purpose of the GLMMs if the results are not reported?

Line 189: There is no record of values/number of nest per year per species/area. Yearly estimates would be helpful in Table 1.

Line 206-243: It is not clear what the importance of these findings are biologically. This study simply checks for synchrony in breeding parameters in different time periods, but it seems as if the authors are searching for significance, with little thought of why one might expect relationships to exist.

Line 250: More explanation could be provided on the importance of examining synchrony in breeding parameters between species, i.e. how this information is of interest biologically or in terms of conservation/species management. Although this study may have identified synchronous fluctuations in breeding parameters between species over time, it is not clear why this is important, particularly as it was only identified in two species. Determining the environmental cause of such synchrony would be interesting, particularly in relation to environmental/climatic changes.

Line 259: Explain why authors think this is a spurious effect.

Line 259: The authors mention the environmental canalisation hypothesis here, but little explanation is given as to what this is and how it applies to this study.

Line 260: Not clear what is meant by “predator guards”. This is more likely to be a local-level effect and is unlikely to affect synchrony in breeding parameters over larger spatial scales.

Line 264: It is not clear why these window sizes were used and whether there was any biological basis for hypothesising that synchrony patterns would be observed at these scales.

Line 268: What evidence is there that synchrony for hatchability has increased in last 20 years?

Line 275: This sentence is highly conjectural, and more clarity is needed in regard to buffering of breeding parameters.

Line 277: A high proportion of ringers (banders) are “citizen scientists” (i.e. they are amateur ornithologists who carry out ringing activities in their spare time and the data are used in scientific studies). Therefore, according to the definition of a “citizen scientist” as a non-professional scientist, ringers/banders are very much in this category.

Line 278: Nevertheless, with appropriate statistical methods, these data are extremely useful for estimating survival (e.g. Freeman and Morgan 1992)

Line 303: It was be useful here to have some description of potential barriers affecting collection of data on some species (e.g. difficulty in accessing nests of some species, local nature protection laws, rarity of species in local areas) and how these barriers may be overcome.

6. PLOS authors have the option to publish the peer review history of their article (what does this mean?). If published, this will include your full peer review and any attached files.

Reviewer #1: **Yes: **Meelyn Mayank Pandit

Reviewer #2: **Yes: **Michael C. Allen

Reviewer #3: **Yes: **Gail Robertson

---

## [Author Response · Author response to Decision Letter 0]

13 Aug 2022

Our responses to the editor and reviewers are below, but can also be found in the Cover Letter and Response to Reviewers_final documents.

Dear Handling Editor,

Thank you for considering our manuscript “Using citizen science to determine if songbird nesting parameters fluctuate in synchrony” for publication and for giving us the opportunity to revise it. We revised the draft to address all comments and believe the revised manuscript is much improved.

Because two reviewers had concerns about us selecting random subsets of bluebird nests to equal the sample size of our chickadee nests, we followed the suggestion of a previous reviewer and used all nests from both species. This reviewer also noted that because we had far more bluebirds than chickadees, when we compared our ecoregions we were actually comparing bluebirds in one ecoregion to bluebirds in the other. Per their suggestion, we removed the ecoregion level from our demographic parameter analyses. This largely resolved the sample size disparities and prevented unnecessary loss of data. Our overall conclusions are the same, and specific results have been updated in the manuscript.

Changes to the manuscript are noted via track changes in the “Revised Manuscript with Track Changes_final” file. Please note that because the authors used multiple computers and Microsoft accounts to make the necessary changes, there are more than two colors of text in our track change edits. In our Response to Reviewers letter, we’ve included a detailed description of how each comment was addressed, with the reviewers’ original comments in italics. 

We hope you find this version to be significantly enhanced. Thank you for your continued consideration.

Sincerely,

Drs. Sara Harrod and Virginie Rolland 

Reviewer’s Comments

Reviewer 1

Thank you for your helpful comments. We rearranged the Introduction to focus more on population synchrony and why it is important to study this phenomenon across multiple populations. We also revised our Discussion to include statement regarding how the lack of synchrony in fledging success (if true) may actually benefit these populations, though it may also be countered by the increased synchrony in hatchability. 

1. Lines 22-25: I would start your abstract introducing what synchrony is and how climate change is affecting it rather than stating the benefits of citizen science. You state that no study has examined synchrony across populations and species but you do not define what synchrony is or why it is important.

a. We agree it would be helpful to define population synchrony and why it is important to study. We revised the Abstract to introduce these ideas. 

2. Lines 30-35: Please state the name (common and scientific) of the species you analyzed. You also need to state that hatching success, hatchability, and fledging success are the breeding parameters you measured; you use the term breeding parameters in the next sentence but you need to state that this term relates to the dependent variables mentioned above.

a. We included our two study species and clarified that hatching success, hatchability, and breeding success are the breeding parameters measured.

3. Lines 36-46: Your conclusions could use some more support; why would breeding parameters be buffered for temporal variation? Are there any other alternative explanations as to why synchrony was not found between populations? You transition into the factors that could have affected your study rather than focusing on what your conclusions say.

a. We added a short explanation of why these breeding parameters may be temporally buffered via the environmental canalization hypothesis, as well as the possibility that there is no population synchrony between these species and populations (lines 36-46). We also state that a lack of synchrony may be beneficial to populations of these study species, though we can’t test for synchrony in other species and populations due to insufficient data (lines 45-48). 

4. Lines 53-56: Your first sentence has a strange transition. You could combine the first two sentences of your introduction to get the point across that long-term and large-scale data are needed to determine how environmental factors and climate change could affect species’ demographic parameters.

a. Thank you for this suggestion. For clarity, we combined the first two sentences. 

5. Line 58: Any songbird examples of the subpopulation or population level not providing adequate information on species responses to change? While the penguin example is good, it is more relatable to have a songbird example since that is what you are studying. Also, the scientific name needs to be in parentheses.

a. Thank you for noting this. We replaced the Adelie Penguin example with that of the House Finch (Haemorhous mexicanus), which according to Breeding Bird Survey data, has experienced population increases in the eastern United States but population declines in many parts of the southwest (lines 58-60). We also added parentheses around all scientific names in the manuscript. 

6. Lines 66-116: I would restructure these paragraphs so that your synchrony paragraphs (66-103) goes before the citizen/community science paragraph (104-116). While the citizen/community science program is important and puts an emphasis on using these dataset in large-scale studies, the main focus of your study is how synchrony affects demographics of species and populations.

a. This is an excellent point! We switched these paragraphs so the emphasis of our paper is on population synchrony of songbird populations. 

7. Lines 78-80: State what specific breeding parameters may be synchronized across populations and similar species.

a. We included a short statement that productivity, or the number of offspring produced per breeding attempt, is one of the breeding parameters that can become synchronized between populations and species.

8. Lines 153-159: Revise sentence to “To examine how breeding parameters between and within two southern cavity-nesting bird species respond to climate change and other changing environmental factors, we used 21 years of community science nesting data from The Cornell Lab of Ornithology’s NestWatch Program.” or something similar.

a. Thank you for the suggestion. For brevity, we combined the first two sentences of this paragraph. 

9. Lines 387-390: You restate your question which is good, but you need to clearly state your conclusions, which I believe begin on line 256.

a. Thank you. We added a short summary of our findings before delving into more detail in the next paragraphs (lines 396-401). 

10. Line 391: State what these negative consequences could be.

a. We stated that the primary negative consequence of increased parameter synchronization is increased risk of extinction. 

11. Lines 418-419: From what I understand hatchability synchrony increased significantly over the past 20 years. You then spend the next few paragraphs discussing gaps in data, examining other species, and the benefits of citizen/community science projects. You need to expand on this result that hatchability synchrony has increased for bluebirds, and what the ramifications are in the context of environmental change and/or climate change.

a. Before addressing this comment, we need note that another reviewer pointed out that by equalizing our sample sizes we unnecessarily lost data. It was also noted that we have far more bluebirds than chickadees, and when we compare the ecoregions we’re actually comparing bluebirds in one ecoregion to bluebirds in the other. We therefore opted to remove the ecoregion level from our demographic parameter analyses. This largely removes the sample size disparities and prevents unnecessary loss of data. Our overall conclusions are the same, and the specific results have been updated in the text and tables. Though we still see an increase in synchrony for hatchability of Atlantic Coast and Central Hardwoods chickadees, we also found a continuously decline in synchrony in fledging success for Atlantic Coast and Central Hardwoods chickadees. Because fledging success of Atlantic Coast and Central Hardwoods chickadees is the only parameter for which we found a significant (α ≤ 0.05) overall correlation in three out of four analyses, this may be a spurious result and we cannot confidently state that these populations are strongly synchronized.

12. Line 89: Avoid starting sentences with “because.” You could use the word “since” instead.

a. Our understanding is that starting a sentence with “because” is grammatically correct as long as the “because” clause is followed by a main clause. The sentence in line 89 is complete with both a subordinate clause (“fecundity often contributes strongly to population growth rates among short-lived species) and a main clause (“increased synchrony in fecundity can increase extinction probability and lessen the potential for demographic rescue”).

13. Lines 179-182: Revise sentences so there is similarity between sentence clauses.

a. We have revised the second clause to be parallel to the first clause.

14. Line 209: After stating the ecoregions, put abbreviations in parentheses after. You use these abbreviations in Table 1; they should be used throughout the rest of the manuscript.

a. We used abbreviations in Table 1 to save space and the caption is close to the table to refer to their meaning. We thought that we should avoid using abbreviations in text to make the reading a little easier. However, your opinion seems different and PLoS One allows the use of abbreviations when they appear at least three times, which was our case with these two ecoregions. Therefore, we changed all instances of Central Hardwoods to CH and Atlantic Coast to AC after defining those terms in line 209. 

15. Line 271: Consider adding an Equation section at the end of the manuscript and having an Equation 1 label here. You also need to state what each of the variables mean in this equation.

a. We are unclear about the Equation section. In reading the guidelines for PLoS One, we couldn’t find such a section. However, we did notice that we should not have included our equation in line with the text because of a hybrid formatting. So, we rephrased the sentence to have the equation appear on a line by itself and we have defined the variables of the equation (lines 273-278). 

Reviewer 2

Thank you for seeing value in our study and for taking the time to review our manuscript. We have reworked the Introduction to highlight implications of synchronous dynamics. We have also developed a clearer prediction at the end of the Introduction. We followed your suggestion in acknowledging the imprecision of our estimates because of the variable annual sample sizes. Though we considered framing our manuscript in the context of climate change, because we did not analyze any correlations between synchrony and climatic variables, we felt this would be outside of the scope of the manuscript. However, we did include discussions of why population synchrony is important in the face of environmental changes such as climate change and habitat loss, and discussed potential ramifications of increased population synchrony. Thanks for your other suggestions. We explain how we addressed each of them below.

1. The abstract is well written but lacks any numerical results. I understand there were few significant effects, but perhaps reporting about the generally high success and low variability observed would be useful.

a. Thank you for this suggestion. We added numerical results to our Abstract and reported our observations regarding high nesting success and low variability (lines 36-37).

2. L41: The phrase “buffered against temporal variability” requires more explanation. This may be where you can insert language about low variability.

a. We removed this phrase and added a short explanation of our results and their implications in the abstract (lines 38-46). 

3. L49-50: suggest deleting this phrase: “to promote the transcription of historic data.”

a. As this recommendation is not at the core of our study and does not change the main message, we have followed your suggestion and deleted this fragment from the abstract.

4. L75-76: suggest changing “occurring simultaneously in intraspecific populations” to “among geographically-separated populations of a single species” or something similar.

a. We have reworded following your suggestion.

5. L78: The phrase “breeding parameters” seems a little vague throughout the ms. Please define it and / or consider using an alternate wording.

a. We replaced “breeding” with “nesting” throughout to be more specific and we provided two examples in parentheses the first time we used the phrase (line 78) as a way to define it.

6. L150: The phrase “Because of continuing threats such as climate change…” does not seem to fit with this sentence and I suggest deleting it. But I do think the motivation for the study needs to be clearly stated further down in the paragraph.

a. In rearranging the Introduction to reframe the focus away from citizen science and more on synchrony, we deleted this phrase and reworded the overall goal in this paragraph (line 153-155).

7. L157-158: I suggest naming the two species here. (And also in the abstract or at least in the keywords.)

a. We provided the names of the two species in the Abstract (lines 30-31) and here (lines 157-158) as suggested.

8. L159-163: You do not state any hypotheses or predictions. I think it would be helpful to state what you expected to find and why it would be important to measure the things you did. These should then be revisited in the Discussion.

a. Thank you for this suggestion! We added our study predictions at the end of the Introduction (lines 166-168), as well as in the Discussion (line 403). 

9. L189: why does the study span such a narrow band of latitude? I’m assuming it was done to limit the variation in temperature? Are there any other major environmental gradients along this east-west transect (e.g., precipitation)?

a. Yes, we wanted to control for variability in temperature associated with a latitudinal gradient. This effect may still be present to an extent through an elevational gradient in the Appalachian Mountains, but this ecoregion was not used in the analyses. Although climatic models project the Atlantic coast to become wetter than more western longitudes used in our study, the current precipitation patterns are comparable, between 40 and 60 inches annually. We have added these clarifications in text (lines 191-192). 

10. L192: citation(s) for non-migratory status?

a. We added the appropriate sources for this sentence.

11. L198: consider replacing “sorted” with “grouped” or some other word.

a. We reworded “sorted nest sites” to “classified nests”.

12. L199: I don’t think the software that you used for this is relevant here.

a. We used that software to overlay the nest locations onto the ecoregion map, but we suppose this step can be done exactly the same with another piece of software. We deleted the software name and its reference here.

13. L205-206: “Though EPA level…” I think this sentence could be improved. For example, what is meant by “more precise”. Do you mean smaller? What does “too small for analysis” mean exactly?

a. In using the phrasing “more precise”, we meant to say that EPA Level III and IV ecoregions are smaller and narrower in geographic scope. Had we used these ecoregions, our nesting data would have been spread across multiple ecoregions, each with less than 10 nests per year. To explain this to the reader, we reworded “more precise” as “smaller in geographic scope” and added a brief explanation of the implications of this for our analyses (lines 207-208). 

14. L208: “We only included data from the Atlantic Coast and Central Hardwoods region…” Then why show all the nests in all the regions on the map? I suggest only showing the nests and region you studied, and explaining how you arrived at that study are in the text.

a. Thank you for this suggestion! We removed all ecoregions except the Atlantic Coast and Central Hardwoods, as well as nests outside these ecoregions. 

15. In general, I think the description of the study area and how it was chosen is a bit disorganized and contains extraneous information. Please try to revise this section to make it more clear and concise.

a. We revised this section to better explain why we chose this study area (i.e., to avoid a latitudinal effect of temperature, to choose sites with comparable annual precipitation, and to avoid a migratory effect of these species; lines 190-192). Although we tried to make this section more concise, we are unclear on what information is considered extraneous and would need further clarification before making further revisions.

16. L217: Please state how many nest records were removed.

a. In total, up to 721 nest records per parameter were removed from our analyses.

17. L221: “probability ≥ one chick fledged” … Please clarify if this was calculated only for nests that had already hatched, or was it for all nests as is typical for “nesting success” measures. That is, was the denominator all nests of the species or only those that reach the nestling stage?

a. This parameter was only calculated for nests in which chicks hatched. We clarified this in our definition of fledging success (line 221). 

18. L221: “probability ≥ one chick fledged” … It is well-established that “naïve” estimates of fledging success (and hatching success too) are biased high due to the inability to discover nests after they have failed. Please consider using an “exposure-days” approach (e.g., logistic exposure modeling) or otherwise assure the reader that this source of bias won’t affect your conclusions. For example, what is the visit interval for the Nest Watch program, and was the visit interval similar among the two species?

a. We agree that the logistic-exposure method is preferred and accounts for nests that fail before they can be discovered, and we have used it in previous studies (Harrod and Green 2018, Harrod and Rolland 2020). Although NestWatch’s Nest Monitoring Manual recommends nests be checked every 3–4 days, many users did not follow this recommendation, and the days between nest visits varied from every day to over two weeks (lines 247-248). Because of these large gaps in nest visits, our annual parameter estimates obtained via logistic-exposure resulted in unrealistic estimates (e.g., all annual estimates for hatching success were 100%). 

b. Additionally, if our estimates are biased high, they are likely biased for both species, and this should not affect the strength of the synchronous relationship between our species-ecoregion pairings (lines 251-253). 

19. L222: “all ecoregions” … This was confusing to me as I was under the impression that all analyses were restricted to just the two ecoregions. I think I understand now, that the “species level” included all the nests on the map, while the other two analyses were restricted to the AC and CH. Please consider clarifying how you present this in the text and perhaps in the map figure as well.

a. Thank you for pointing this out! We agree the wording is confusing and have changed this statement from “one species across all ecoregions” to “one species across the CH and AC ecoregions” (lines 285-286). 

20. L226: “Due to low annual…” This is repeated information from L125-126.

a. Because our revised manuscript no longer includes the ecoregion level in our demographic parameter analyses, we removed this sentence.

21. L231: “…by randomly selecting a subset of bluebird nests…” Does this mean that for each year, the same number of bluebird and chickadee nests were included in the analysis? Or just the same total sample size? Please clarify.

a. We ensured that the same number of bluebird and chickadee nests were included in each analysis. For example, if we had 1,000 bluebird nests and 500 chickadee nests, we took a random subset of 500 bluebird nests before proceeding with the analyses. However, please see our response to the next comment. 

22. L231: While I see the value in equaling the sample sizes to ensure compatibility of the datasets, it also discards information and introduces greater uncertainty in the bluebird estimates. I think the authors should consider a bootstrapping-type randomization approach that repeats this process many times to avoid discarding information. At the very least, repeating the random selection and analysis say 5 times and reporting whether or not the conclusions changed would do a lot to alleviate this concern. The primary results could remain as is, but with the addition of a statement that repeating the process did not influence conclusions.

a. We initially used the equal sample size approach because of the comments of a previous reviewer. Although we tried repeating the randomized selection and analysis to avoid losing information, the results changed among simulations. 

b. Because this same reviewer also pointed out that since we have far more bluebirds than chickadees, and when we compare the ecoregions we’re actually comparing bluebirds in one ecoregion to bluebirds in the other, we opted to remove the ecoregion level from our demographic parameter analyses. This largely removes the sample size disparities and prevents unnecessary loss of data. Our overall conclusions are the same, and the specific results have been updated in the text and tables. 

23. Table 1. Can you report the min and max annual sample size of nests across years in a new column (or columns) here? The only information I see about annual sample size in the methods is a rule to ensure at least 10 nests per year. But then you mention later that there was one year of no data in the Central Hardwoods. Please clarify. I’m guessing Central Hardwoods has at least 10 nests per year except for the one year which had none? Summary statistics would be helpful.

a. We apologize for the confusion. In our manuscript we intended to state that we ensured our samples had an average of at least 10 nests per year but left out the words “average of”. We corrected this oversight in the Methods. 

b. We revised Table 1 to include the annual range of nests across years. We also noted when a particular level was missing one year of data.

24. Table 1. Please clarify how these estimates and SE were produced. Are they model predictions or simple averages across years?

a. These estimates were calculated by obtaining the predicted annual estimates from the GLMM models, then averaging these estimates and calculating the standard error. For clarity we added a statement in Table 1’s caption explaining this process. 

25. L241: What does “at all levels” mean here? Please clarify. Also please clarify the model structure: did you get the estimates by including year as a categorical variable in the model?

a. “All levels” means the species, ecoregion (now removed from the demographic parameters analyses of this revised manuscript), and species-ecoregion levels. For clarity, we rephrased this to say “we estimate annual parameters at the species and species-ecoregion levels”. 

b. In all models we obtained our estimates by including year as a factor variable. We clarify this in line 241 of the text. 

26. L254-255: What is the definition of “nest site” and “nest location” here? Is it the same nest box? Or the same geographic coordinates? Please clarify. How common were “multiple-nesting-attempt” locations in the data set? On average how many attempts per location?

a. We apologize for the confusion; both “nest site” and “nest location” were intended to refer to the coordinates of the nest. To avoid confusion, we rephrased lines 253-255 as “To account for pseudoreplication among nest locations, we included the nest location as a random variable”. Multiple-nesting attempts at the same location were very common across the data set, with some locations having over 100 attempts over the course of the study period (1998–2018). 

27. L253-262: I think including nest location as a random-effect is advisable whether or not it performs better based on AIC. Multiple nesting attempts from the same nest box, for example, seems like a clear example of non-independence. Multiple attempts from the same box within the same year are even less likely to be independent. As this only affects one model in your analyses, I think using a random-effect for all and dropping the language about model selection from this paragraph would streamline things and make for a more consistent and appropriate analysis.

a. Thank you for this suggestion. For consistency across our analyses, we included nest location as a random effect (lines 255-256). 

28. L249: please clarify the “family” of GLM you used to assess these models. I am assuming it was Binomial for the binary data. However, was a Gaussian or Binomial model used for the proportion data? The best way to approach the proportion data, in my opinion, is to use a Binomial GLM with the “weights” argument specifying the total number of eggs in each nest. Here is an example from an online posting: https://stackoverflow.com/questions/50078497/weighted-logistic-regression-in-r

a. We used a binomial error distribution for all models, whether the response was binary or proportion data. The confusion might be in that we didn’t use percentages for hatchability. If we had, we agree that a weighted regression would have been appropriate. The problem with percentages is that the typical arcsine transformation to meet normality doesn’t always work. Instead, we used the original counts: number of hatched eggs (success) and unhatched eggs (failure) per nest. We combined those two counts into one response variable. In R, the code would look like glm(cbind(success,failure)~predictors,binomial,data). We attempted to clarify this in text (lines 244-245).

29. L273: Sigma and mu are not defined here.

a. We have now defined them after the equation as well as the maximum value (line 274).

30. L271: Why not report the coefficient of variation (sd/mean) for this (instead of, or in addition to, the other measure)? It is similarly standardized and might be easier for the general reader to interpret.

a. Thank you for this suggestion! For ease of interpretation, we calculated the coefficient of variation of our parameters (lines 275-278) and added their values to Table 1.

31. L289-291: “For example, to determine if hatching success…” I recommend modifying and clarifying this example. On first read, it doesn’t seem appropriate to estimate a correlation over only 3 or 4 years of data. But if this is indeed how the technique works, please expand on this a little more to convince the reader it is appropriate. Are the sliding-window estimates then averaged? Or are they plotted to look for time periods of higher correlation? This is an interesting approach and I think other readers would also benefit from a little more detail about how it is evaluated. Please clarify. Why not look for correlation over the entire time series as well, similar to other population spatial synchrony work?

a. We apologize for the confusion; we intended for this to be a hypothetical example to introduce readers to the concept of the sliding window correlation. A three-year window would be very small, so to avoid confusion we substituted this example for one with a five-year window (lines 290-291).

b. We also added a brief description of how we determined whether our sliding-window correlation estimates were significant (lines 292-293).

c. Finally, we did run our correlation analyses over the entire time series at each level (line 297). 

32. L300: “combinations with …” Please clarify: combinations of what?

a. We apologize for the confusion. We intended to state that we conducted sliding-window correlations for combinations of species and ecoregions, and for clarity specified that we ran our correlation tests at the species and ecoregion levels. 

33. L307: Suggest changing “…Central Hardwoods.” to “…Central Hardwoods regions.”

a. Because we defined our ecoregions in Table 1 and PLoS One allows the use of abbreviations when they appear at least three times (which was our case with these two ecoregions), we changed all instances of Central Hardwoods to CH and Atlantic Coast to AC after defining those terms in lines 208-209. 

34. L325: Again, I think most readers could use some guidance in interpreting “relative variance” (e.g., what is a high value?). Consider using CV.

a. We defined what constitutes a low and high value in lines 279-280. Please see our response to comment 30 regarding CV.

35. L335-353: I’m not very familiar with the sliding-window approach, but reports of a few short intervals that were significantly correlated across time do not seem very meaningful to me. I think it would be preferable to report measures of overall time series cross-correlation. Then, where none was found, report that some scattered short intervals were correlated and refer readers to the figure instead of presenting each one. It is easy to get confused by reading a report of each significant “window” without getting the overall context that in general there were few meaningful patterns. Perhaps a table would help to organize these results and allow for a more streamlined section.

a. We agree that reports of a few short significantly correlated intervals do not appear meaningful, which we discuss in lines 424-246 and lines 427-428 of the Discussion. As stated in line 298, we did measure the overall correlation coefficients for each parameter, but the only overall correlation coefficient that was significant was for fledging success of Atlantic Coast and Central Hardwoods chickadees (lines 336-367, 421-424).

b. To help readers understand the overall context of our results, we added statements that we did not find any significant correlations over the entire time series for hatching success or hatchability (lines 338-339 and 346-347, respectively) before delving into the period for which parameters were correlated. 

36. L390: “In this study, using 21 years of…” Often, the first paragraph of the Discussion includes a concise high-level summary of your results. But if you choose not to go that route (which is fine with me), this sentence seems out of place as it simply states what you studied without providing any results.

a. Thank you for this suggestion! After discussing our results, we intended for this paragraph to re-focus and remind readers of the background of our study, why we studied spatial synchrony in nesting parameters, and the importance of studies such as this. However, to better the flow between paragraphs, we provided a short summary statement of our results at the end of this paragraph (lines 399-404) before the next paragraph delves more into individual results and their implications. 

37. L405-411: I would define the term “canalized” again here in the Discussion if you think it is important to include. I think many readers would benefit from a reminder of what it is exactly, and potentially some more background about the hypothesis you reference.

a. We clarified in line 396 that when parameters are not canalized they fluctuate strongly and briefly redefined the environmental canalization hypothesis in lines 409-410.

38. In general, I think the Discussion lacks in-depth speculation about the causes and the implications of having synchronous breeding parameters.

a. Because we did not find many instances of population synchrony in our parameters (using sliding-window correlations and the overall correlations), we did not feel it appropriate to delve deeply into the causes of strongly synchronous nesting parameters on these populations. However, our Introduction (lines 71-74 and 80-82) discusses the two possible mechanisms for which this synchrony can occur, and the Introduction (lines 91-92) and Discussion (lines 394-396) discuss possible consequences of having strongly synchronized parameters. We also added a speculative scenario to the Discussion regarding specific consequences that could be seen from our results (lines 444-448). 

39. L408: Why is the effect potentially spurious? Also, I thought no time series were found to exhibit a linear time trend?

a. We believe this effect was potentially spurious because out of all 15 nesting parameter analyses at the species and ecoregion levels, only one (species-level bluebird hatchability) was found to have a significant (α ≤ 0.05) temporal trend. 

b. To answer the second part of your question, although species-level bluebird hatchability exhibited a significant trend, this estimate was obtained by analyzing combined nesting data from both ecoregions. Had we compared species-level bluebird hatchability to species-level chickadee hatchability, we would have needed to detrend these data. In our sliding-window correlation however, we looked for correlations between the annual estimates from AC and CH bluebirds, which did not require detrending. 

40. L419: “but see Figs 2a…” This should refer to Fig. 3 instead.

a. Thank you for pointing out this error! We have corrected this sentence to reference Figure 3.

41. L422-242: Again, I’m confused as I thought no time series were found to have a trend.

a. We apologize for the confusion in our wording. In calculating our nesting parameter estimates at the species-ecoregion level, we did not find a significant time-series trend for any parameter. However, we did find an overall significant correlation in fledging success between AC and CH chickadees with our sliding-window correlations. To avoid confusing readers, we changed “this is the only parameter for which we found a consistent trend” to “this is the only parameter for which we found a significant overall correlation.”

42. L427-428: One other major factor could have led to non-significant correlations: uncertainty in the breeding parameter estimates themselves which were based on variable amounts of data (n = ~ 10 to ? per year). I realize that it would be a considerable computational challenge to fully address parameter uncertainty in this analysis, but it should at least be discussed. If you found clear patterns, that would argue that parameter uncertainty could safely be ignored. The fact that you did not find clear patterns suggests that it may be important to consider in future analyses. You touch on this in L279-280, but I think it deserves more attention.

a. This is a good point. However, although the number of nests varied among years (n = 1 to 283 per year), the mean estimates were associated with small relative variance, but this was for the whole period. The annual estimates that we used for the sliding-window correlation may have been imprecise in some years with few nests. We added a sentence in the Discussion (lines 438-440) about this potential source of bias.

43. L436-437: Please reword: “…is enough to wonder if this result is spurious,…”

a. We have reworded to “may be a spurious result,...”

44. L466: Missing an “and” before “Carolina Chickadees”

a. We are unclear about this suggestion because Carolina Chickadees are not the last species in the list.

45. L459-475: This paragraph discusses the need for more citizen science nest monitoring data without tying it back into how it would benefit our understanding of the synchrony of breeding parameters, which is the focus of the next (very short) paragraph. I suggest merging these two into one concise paragraph, and spending more time on the research implications and less on the various programs and species.

a. We agree that merging these paragraphs makes sense and have combined them. However, we also feel that our discussion of specific programs and species is necessary because it provides an overview of the progress that has been made in recruiting new citizen scientists and transcribing historic data. We hope that by combining these two paragraphs, it becomes clear to the reader that with more data we can re-test for synchrony in our study and other studies, and determine whether the lack of synchrony is due to data shortcomings or a true absence of synchrony.

46. One additional topic of discussion to consider: cavity nesting species are known to have higher nesting success (lower predation rates) than non-cavity nesting species. Could this be one reason you found relatively low interannual variability, and might you find different results with non-cavity nesting species?

a. This is an excellent point! We acknowledge that cavity-nesters tend to have higher nesting success than their non-cavity-nesting counterparts, and this could be a reason for lower RV and CV values. As stated in our Methods, we chose cavity-nesting species because we assumed they would have greater sample sizes since observers would not need to search for nests. We added a statement (lines 411-412) acknowledging that cavity-nesting birds have higher nesting success, and this could have contributed to our lower RV and CV values. 

Reviewer 3

Thank you for your time providing us with a thorough review and suggestions. We have clarified that we chose breeding parameters because those contribute highly to population growth rates of short-lived species and that we are not aware of a citizen science program that collects survival data but that survival would be worth examining. Typically, to study survival, one needs banding data and many citizen scientists do not hold such permits. The Bird Banding Lab who collects banding data from permitted banders only consistently collects data on first-time banded birds, but banders are not mandated to provide recapture/resight data, which are necessary to estimate survival probabilities. Besides, the Bird Banding Lab is not allowed to share these data and can only email relevant banders to contact the researchers if they’re willing to share data.

For clarity we also provided a definition of citizen science. We were unaware of the paper by Robertson et al. (2015) and therefore amended our Introduction to state that few studies have used citizen science data to assess fluctuations in demographic parameters and cited this source. Regarding the structure of our Discussion, we focused less on the citizen science data discussion at the request of another reviewers. Because of this restructuring, we revised our third study objective to state that we discussed some benefits and constraints of using citizen science data. 

We agree with your suggestions that examining the causes of synchrony between species would increase the impact of the study. However, one major challenge is that we’d need to find synchrony before we could examine causes of synchrony. Also, adding this interesting aspect would considerably lengthen the paper because it would require providing specific hypotheses about various climatic variables, more analytical methods (e.g., how to obtain the variables for each ecoregion and across), and of course more results and discussion. Because of these issues and because it’s out of scope of our paper, we decided not to include this examination. 

Our GLMMs were used to estimate annual estimates of the nesting parameters. We then used those estimates to check for temporal trends, temporal variability, and correlations. We clarified this in our Methods. The Discussion is not about estimates specifically, although we do highlight how high the nesting parameter estimates were; instead, it focuses on about temporal variability and synchrony.

1. Lines 29-36: The authors do not make it clear why examining synchrony across these species is important. The abstract begins by focusing on the utility of citizen science, but fails to link this premise with the study aim of examining synchrony in demographics two cavity-nesting species. 

a. We apologize for this oversight. We added a section to the Abstract (lines 22-25) explaining why, in the context of global climate change, it is important to determine whether demographic parameters are synchronized across inter- and intraspecific populations, and how increased synchronization can negatively impact these populations.

2. Lines 30-31: Not clear here why the authors aimed to compare these two species - were the two species chosen simply because data were available or was there an underlying biological hypothesis driving this study? If so, make explicit here.

a. Your first assumption is correct; we chose these species because of the availability of data, and because they are widespread and have similar life histories. We added a statement to this effect in the Abstract (line 32).

3. Line 40-41: “Breeding parameters could be buffered against temporal variability”; not clear what this phrase means.

a. To avoid lengthening the Abstract with an explanation, we removed this phrase and stated that our parameters showed little variability (line 37). 

4. Lines 106-109: The authors need to better define "large-scale scientific studies" and "citizen science programs". There may be some overlap between these (e.g. scientific studies may rely on data collected by volunteer amateur ornithologists, so not clear under these definitions whether these should be defined as “large-scale scientific studies” or “citizen science programs”)

a. Although there can be overlap between these definitions, not all citizen science programs are large-scale. For the purpose of this manuscript we define traditional scientific studies as “studies in which data are collected over large geographical scales by trained researchers with specialized education and experience in their field of study”, and citizen science programs as “programs in which individuals from any background collect data over varying geographical scales for use by professional scientists” (lines 106-110).

5. Lines 114-116: It's not clear here what the authors mean by "citizen science data". Many studies have made use of data collected by ringers or bird watchers who were not directly involved in the study, but who collected data that were later used. For example, many studies have been published using long-term monitoring, collected by amateurs or reserve wardens and later analysed to examine synchrony among populations or species (see Robertson et al. 2015, Lahoz-Monfort et al. 2017).

a. Please see our responses at the start of this letter, our response to the previous comment, and our definition of “citizen science programs” in the manuscript (lines 109-110). 

6. Line 71-72: Could be clearer regarding which phenomenon the authors are referring to here.

a. We clarified that we are referring to the phenomenon of synchronous fluctuations (line 72).

7. Line 77: Add reference after “it can also be applied to closely related species”

a. We added the appropriate reference to this statement. 

8. Line 78: Add reference after “and species with similar life histories”

a. We added the appropriate reference to this statement. 

9. Line 78-80: Explain here what factors are driving this synchrony and give examples of how breeding parameters fluctuate across populations and species.

a. The same factors that drive synchronous population fluctuations, which are discussed in line 72, (i.e., dispersal and the Moran effect) can drive synchronous fluctuations in nesting parameters. For clarity, we provided examples of specific factors identified in a study of synchronous productivity among five seabird species (Lahoz-Monfort et al. 2013; lines 80-82). 

10. Line 84: In Figure 1 legend the authors could explain how data were simulated in this example.

a. The data presented in Figure 1 are entirely hypothetical, which is why we did not provide specific years for the x axis or indicate what parameter was considered on the y axis. We just made up data such that the parameters would be perfectly positively correlated (r = 1), perfectly negatively correlated (r = -1) or not correlated at all (r = 0) to illustrate positive, negative, and no synchrony in parameters, respectively. If the values on the y axis are what makes the interpretation of this figure’s purpose confusing, we could certainly remove them.

11. Lines 89-90: Fecundity is defined as number of young produced per female (see Nagy and Holmes 2004, Journal of Avian Biology 35: 487-491) rather than number of fledged female offspring as stated here.

a. We found multiple definitions of fecundity across the literature. Zink (2000) defines fecundity as the total number of eggs produced by a female, whereas Hill et al. (2010) defines it as the mean clutch size per female and van de Pol (2010) estimated fecundity as the number of fledglings produced by a breeding pair in a year multiplied by the sex ratio.

b. We chose to go with the definition provided by Saether and Bakke (2000), which has also been used by Frei et al. (2015), Donovan et al. (2008), and Etterson et al. (2011). 

12. Lines 91-92: Explain how “increased synchrony in fecundity can increase extinction probability and lessen the potential for demographic rescue”

a. The example in the following sentence (lines 92-99) is meant to explain how increased synchronization in fecundity can increase risk of population extinction and lessen the likelihood of demographic rescue. For clarity, we added that nesting parameters contribute to fecundity (line 93), which we previously stated (lines 90-91) contributes strongly to population growth among short-lived species. 

13. Lines 93-94: This phrase is confusing, the authors should be clear what they mean here. Intraspecific is defined as “produced, occurring, or existing within a species or between individuals of a single species”, therefore the phrase “intraspecific passerine populations” doesn't make much sense.

a. We understand that “intraspecific” is often used in the context of competition, but it has also been used in the context of genetics and demography. In our example, we used the etymologic meaning of intraspecific that is “within a species” to say two populations of the same species. We clarified our intent by adding “(i.e., two populations of the same species)” in our manuscript (line 94). 

14. Lines 150-170: This last paragraph attempts to explain why this study was carried out, the specific aims of the project, and how the study contributes to ecology and conservation. I think the authors could make this section clearer - the aims are not well explained and it is not clear how they relate to the text above. What is meant by “two spatial scales” and why is it important? Why did the authors choose these breeding parameters? These are questions that should be addressed here.

a. We apologize for the lack of clarity. We intended to say that we estimated the nesting parameters of bluebird and chickadee populations between and within regions, and assessed population synchrony between and within species. We corrected this mistake and briefly defined these levels in the Methods (lines 286-287).

b. It is important to state why we chose our breeding parameters, but this belongs in the Methods section. To be more specific, we changed “breeding parameters” to “nesting parameters” and stated that we estimated these parameters because they were available through our data (lines 224-226). 

15. Lines 184-185: “allowing us to assess breeding parameter synchrony across a large spatial scale.” Explain why is this important.

a. In our first paragraph (lines 54-58) we explain why spatially large-scale studies are important. To briefly tie this back together, we added a statement explaining that assessing synchrony across a large spatial scale would allow us to “draw more informative conclusions at the species level” (line 185). 

16. Line 189: This is a large area. You might expect populations closer together to exhibit more synchronous breeding behaviour compared with species further apart in this range. Did the authors test this hypothesis? This appear to be the case according to text below, but is not made clear in the study objectives in the Introduction.

a. If the question is about species closer vs. further, we could not test for this because while we checked for synchrony between regions (i.e., further), we didn't check for synchrony within a region (i.e., closer) for a given species. This was outside the scope of our objectives. If the question is about populations closer vs. further, then it depends whether populations are of the same or different species. We did check for interspecific synchrony within a region (i.e., closer), but we didn't have enough nests to check for interspecific synchrony between regions (i.e., further). We clarified the scales of comparisons in the paragraph of our objectives (lines 160-161).

17. Line 194: Perhaps the authors could colour points to show number of nests available per year in Figure 2.

a. This is a great suggestion, but we fear that with 21 years of data (thus 21 colors), incorporating this suggestion would make the figure visually overwhelming to readers. We instead provided the annual range for the number of nests per year in Table 1. 

18. Line 202-203: “MBJV ecoregions were defined using avian ecological data”. Describe how this was done and how this method differs from EPA Level II ecoregions.

a. The Migratory Bird Joint Ventures ecoregions are similar to the EPA’s Level II ecoregions, but were developed by the North American Bird Conservation Initiative (NABCI). Ecoregions are delineated not just by their general biotic (plant communities and wildlife) and abiotic (soil, temperature, and precipitation) features, but also by the similarity of bird communities and resource management issues. We added a statement to this effect in lines 203-204. 

19. Lines 209-210: It looks from Figure 2 that there would have been enough data for analysis in the Appalachian mountain region. The authors could explain here why this was not possible, even for a smaller number of years.

a. We excluded the Appalachian Mountains because there were multiple years with an average of less than 10 nest records per year (lines 210). Per the suggestion of another reviewer, we removed, from the map, all nests not used in our analyses (e.g., those in the Appalachian Mountains). 

20. Line 217: Report how many nests per area/year/species contained errors.

a. We believe reporting the number of nest removed per area/year/species would lengthen the manuscript unnecessarily with such level of details as it would correspond to 2 regions x 2 species x 21 years = 84 values of number of nests removed to report. We did add that up to 721 nest records per parameter were removed because of errors (line 217). 

21. Line 219: More often “percentage of eggs which hatch in a clutch” is used to define hatching success. Not clear how the authors calculated the probability that at least one egg hatched. If this is a metric calculated from data, why would a probability be needed? Wouldn't proportion/percentage be correct?

a. We understand that, as with fecundity, varying definitions of hatching success exist. Though many papers define hatching success as the percentage of eggs hatched in a clutch, other papers (Smith et al. 2009, Durant et al. 2010) use the same definition as our manuscript. The idea is that hatching is “successful” is at least one of the eggs in the clutch hatched. To better explain how we calculated the probability that at least one egg hatched, we added an explanation in lines 249-250 that we created generalized linear mixed models with binomial error distributions, and in lines 242-243 that our response variable was binary (with “1” indicating at least 1 egg hatched and “0” indicating no eggs hatched). Examining hatching success as we defined it separately from hatchability can give a more precise idea of what happens at what stage.

22. Line 220: What is meant by a “successful” nest?

a. In the case of hatchability, a successful nest is a nest where at least one egg hatches (i.e., nests in which no eggs hatched are excluded from this analysis). For clarity, we rephrased “proportion of eggs hatched in a successful nest” to “proportion of eggs hatched in a nest in which ≥ one egg hatched” (line 220). 

23. Lines 230-232: This would reduce the size of your dataset unnecessarily. Perhaps accounting for differences in sample sizes in your statistical analysis would have preserved sample size in this case.

a. We initially used the equal sample size approach because of the comments of a previous reviewer. Because this same reviewer also pointed out that since we have far more bluebirds than chickadees, and when we compare the ecoregions we’re actually comparing bluebirds in one ecoregion to bluebirds in the other, we opted to remove the ecoregion level from our analyses. This largely removes the sample size disparities and prevents unnecessary loss of data. Our overall conclusions are the same, and the specific results have been updated in the text and tables. 

24. Table 1: Define what is meant by “relative variance” in the table heading. Also, some numbers are overlapping in Table 1

a. We explained in the table heading that relative variance (RV) and the coefficient of variation (CV) are measures of variability of our parameter estimates. The overlapping numbers seem to be an error with the conversion of Word to PDF during the submission process as 145 and 146 are line numbers.

25. Line 249: It isn't clear here what the authors have done. What was the GLMM used for? What response and explanatory variables were included in the model and what was the reason for doing this? If the purpose is to compare breeding parameters between species/area/year these variables should have been included as explanatory variables. It isn't clear what is meant by “estimated annual parameters” given that in lines 217-221 it is stated that these were calculated from the data.

a. We apologize for the confusion. We explained in lines 249-250 that we used GLMM models to obtain annual estimates of all three nesting parameters. The response variables for hatching and fledging success were binary, whereas hatchability had a proportion distribution and used a two-vector response variable (one vector for the number of hatched eggs per nest and the other for the number of unhatched eggs per nest). For all three parameters, “year” was included as an explanatory factor variable to obtain the annual parameter estimates.

b. To avoid confusion, we rephrased the early wording from “estimated annual parameters” of hatching success, hatchability, and fledging success to “considered” hatching success, hatchability, and fledging success (line 219).

26. Line 258: How can you compare a linear mixed model in which one of the models included no random effect? Presumably a different model was used - state here what this was.

a. Just as models with and without fixed effects can be compared with Akaike’s Information Criterion, so can models with and without random effects. In this case, models had the same response variable and no fixed variables; the only difference in models was the presence/absence of the random effect. 

b. However, for consistency across our analyses, and following another reviewer’s suggestion, we opted to include nest location as a random effect in all our models (lines 256-257).

27. Lines 262: How many years of data were available? State here.

a. We previously stated in line 156 that 21 years of data were available. Including this information here would be extraneous since the purpose is to report that year was treated as a fixed factor.

28. Line 264: Explain what these linear models were and how they were used in this example.

a. Thank you for this suggestion. We specified in lines 264-265 that the annual estimates (used as the response variable) were examined against years (used as a categorical, explanatory variable). We evaluated the temporal trend looking at the P-value of the slope to determine if it’s significant at α = 0.05.

29. Line 290: This is a short time series over which to determine whether there is a correlation. Did the authors check to see whether there were enough data points to determine whether there was a significant correlation?

a. The time series provided here was provided as an example to help the reader understand how sliding-window correlations work. We did not use three-year windows in any of our analyses. To avoid confusion, we reworded this example using a five-year window, which was used in previous synchrony studies (lines 290-292).

30. Line 295: Explain why these window sizes were used.

a. We added a brief explanation (lines 296-297) stating that 5-years is the smallest meaningful window size and that previous studies (e.g., Durant et al. 2004) found that the same patterns in synchrony were visible with five-, seven-, and nine-year windows. 

31. Line 307: Report numbers of nests for each species/area/year here.

a. Because of the multiple combinations of species, ecoregion, and year, this (just like the number of nests removed because of errors) would result in an unnecessary and extraneous amount of information. This paragraph states the percentage of observations per species and ecoregion, and Table 1 lists the range of nests per year. 

32. Line 312: It is not clear where the results of the GLMMs are in the Results section. What was the purpose of the GLMMs if the results are not reported?

a. As the GLMMs were used to estimate nesting parameters, we reported these results under the subsection “Nesting parameters.” To avoid redundancy, we chose to not repeat the exact values that are listed in Table 1 and instead focused on overall results (e.g., hatching success was highest for AC and CH bluebirds). For clarity, we added a statement specifying that hatching success, hatchability, and fledging success estimates were obtained from our generalized linear mixed models (line 313). 

33. Line 189: There is no record of values/number of nest per year per species/area. Yearly estimates would be helpful in Table 1.

a. Please see our response to comment 31.

34. Line 337-381: It is not clear what the importance of these findings are biologically. This study simply checks for synchrony in breeding parameters in different time periods, but it seems as if the authors are searching for significance, with little thought of why one might expect relationships to exist.

a. We did not feel it appropriate to discussion the significance of our findings in the Results, as this is usually reserved for the Discussion section. We do discuss why synchrony is important to study in the Introduction (lines 91-92 and lines 168-170) and Discussion (lines 389-390 and 394-399), and we discussion implications and potential reasons for the lack of synchrony in our study (lines 427-432). However, we did add a correlation for the overall period so that we also have a bigger picture for synchrony. In the case of no overall synchrony, looking at smaller periods allows us to see if synchrony existed and disappeared and there is no reason to expect synchrony to be permanent.

35. Line 388: More explanation could be provided on the importance of examining synchrony in breeding parameters between species, i.e. how this information is of interest biologically or in terms of conservation/species management. Although this study may have identified synchronous fluctuations in breeding parameters between species over time, it is not clear why this is important, particularly as it was only identified in two species. Determining the environmental cause of such synchrony would be interesting, particularly in relation to environmental/climatic changes.

a. We added a statement reiterating that increased synchrony in parameters across species and populations can increase the risk of extinction and (in the case of intraspecific populations) lower the chances of demographic rescue (lines 393-394). We also added a sentence in the Discussion (lines 396-399) explaining that detecting interspecific synchrony can help researchers predict which species are likely to experience simultaneous declines in a demographic parameter. 

b. We agree that determining the environmental cause(s) of synchrony is important, and mentioned two potential mechanisms in the Introduction (lines 72-74). In our specific study, we found little evidence of synchrony and therefore cannot speculate as to its causes. 

36. Line 408: Explain why authors think this is a spurious effect.

a. We believe this effect was potentially spurious because out of all 15 nesting parameter analyses at the species and ecoregion levels, only one (species-level bluebird hatchability) was found to have a significant (p ≤ 0.05) temporal trend. 

37. Line 409: The authors mention the environmental canalisation hypothesis here, but little explanation is given as to what this is and how it applies to this study.

a. We added a brief summary of this hypothesis to the end of the sentence in question (lines 409-410). 

38. Line 412: Not clear what is meant by “predator guards”. This is more likely to be a local-level effect and is unlikely to affect synchrony in breeding parameters over larger spatial scales.

a. We apologize for the confusion and listed examples of common predator guards used to protect nest boxes (lines 412-413). Predator guards have been demonstrated to improve nest survival 6.7% at the national level (Bailey and Bonter 2017), and because we do not know how many nests were protected, we cannot rule out the possibility that they buffered our annual parameter estimates against interannual fluctuations. 

39. Line 417: It is not clear why these window sizes were used and whether there was any biological basis for hypothesising that synchrony patterns would be observed at these scales.

a. Please see our response to comment 30.

40. Line 421-422: What evidence is there that synchrony for hatchability has increased in last 20 years?

a. Although this claim is still true based on Figures 3a and 3b, we updated this part of the Discussion to reflect the new results after redoing our analyses (see our responses at the start of this letter). In this revised manuscript, the contrast we are highlighting is for fledging success; synchrony in this parameter declined over the last 20 years. This is referring to Figures 3b and 3c, which we presented in the Results (lines 372-375). We did not refer to those figures in the Discussion as it is generally not recommend to refer to figures in this part of the manuscript. However, we did clarify that the contrast we made was with the correlation for the overall period (line 243). 

41. Line 430-431: This sentence is highly conjectural, and more clarity is needed in regard to buffering of breeding parameters.

a. We replaced “buffered” with “canalized” and added a citation for our statement that survival tends to be less canalized for short-lived passerines (lines 431-432). We agree that our statement regarding synchronization of survival between and within species is conjectural, but feel it is worth mentioning as a possible explanation for why we found little synchrony in nesting parameters. .

42. Line 432: A high proportion of ringers (banders) are “citizen scientists” (i.e. they are amateur ornithologists who carry out ringing activities in their spare time and the data are used in scientific studies). Therefore, according to the definition of a “citizen scientist” as a non-professional scientist, ringers/banders are very much in this category.

a. We agree but the issue is that citizen scientists are not all banders and we are not aware of a citizen science program that collects capture-recapture/resight data. We clarified this last point in line 434. 

43. Line 435: Nevertheless, with appropriate statistical methods, these data are extremely useful for estimating survival (e.g. Freeman and Morgan 1992)

a. We agree the importance of banding and recovery studies cannot be overstated. However, recovery rates are often low, and a very large number of birds must often be banded to achieve modest recovery rates necessary for such analyses. 

44. Line 472: It was be useful here to have some description of potential barriers affecting collection of data on some species (e.g. difficulty in accessing nests of some species, local nature protection laws, rarity of species in local areas) and how these barriers may be overcome.

a. This is a great suggestion! We added an example of a potential barrier and ways in which programs can overcome these obstacles (lines 471-474).

---

## [Decision Letter · Decision Letter 1]

13 Sep 2022

PONE-D-22-01496R1Using citizen science to determine if songbird nesting parameters fluctuate in synchronyPLOS ONE

Dear Dr. Harrod,

Thank you for submitting your revised manuscript to PLOS ONE. Two of the previous reviewers looked at the revision. One felt that all of his/her comments were addressed adequately, while the other raised a few issues that still need to be attended to.  Please prepare a revision that addresses these comments.

We look forward to receiving your revised manuscript.

Sincerely,

Charles R. Brown, Ph.D.

Academic Editor

PLOS ONE

Journal Requirements:

Reviewers' comments:

Reviewer's Responses to Questions

**Comments to the Author**

1. If the authors have adequately addressed your comments raised in a previous round of review and you feel that this manuscript is now acceptable for publication, you may indicate that here to bypass the “Comments to the Author” section, enter your conflict of interest statement in the “Confidential to Editor” section, and submit your "Accept" recommendation.

Reviewer #1: All comments have been addressed

Reviewer #2: (No Response)

2. Is the manuscript technically sound, and do the data support the conclusions?

Reviewer #1: Yes

Reviewer #2: Yes

3. Has the statistical analysis been performed appropriately and rigorously? 

Reviewer #1: Yes

Reviewer #2: Yes

4. Have the authors made all data underlying the findings in their manuscript fully available?

Reviewer #1: Yes

Reviewer #2: Yes

5. Is the manuscript presented in an intelligible fashion and written in standard English?

Reviewer #1: Yes

Reviewer #2: No

6. Review Comments to the Author

Reviewer #1: The authors have addressed my comments to the best of their abilities and have conducted an interesting experiment. I look forward to seeing this manuscript in publication.

Reviewer #2: General comments:

This manuscript is much improved over the previous version. I commend the authors for their thoughtful revisions. I find the analysis to be sound, interesting, and generally well presented. However, the manuscript still contains several instances of imprecise or unclear language, in my opinion, especially in the Introduction and Discussion. I provide suggestions for those areas in the detailed comments below. Addressing these, I believe, is needed to make the meaning of the (interesting!) results and their potential implications clearer. A careful readthrough for typos is also necessary, as I discovered a few during my last reading (also noted in the detailed comments). With a little more polishing, I believe this manuscript will make a valuable contribution to our understanding of population synchrony among and within species, as well as the challenges faced in measuring these phenomena.

Specific comments:

Abstract

L24: I suggest deleting the phrase “inter- and intraspecific”. Or else changing to “inter- and intraspecific interactions among wildlife populations”. Otherwise, the meaning is unclear.

L27: typo: double comma

L39: “…study species may be resilient to climate change”: Can you insert somewhere earlier in the abstract that weather/climate is a common driver of spatial synchrony in population dynamics? That would make this statement seem more believable, connecting the dots between climate change and population synchrony.

Altogether, a much improved Abstract.

Introduction

L48-50: This sentence requires refinement, perhaps as follows: “Studies focusing on sub-populations and smaller regions are common, but are inadequate for understanding population dynamics across a species' range.”

L52: typo: double period after “United States declined [5]”

L53-55: Most widespread populations exhibit both areas of high and low spatial synchrony within populations (e.g., Allen et al. “Mapping spatial synchrony…”, Conservation Bio, 2021). I suggest revising to something like the following: “...likely vary among populations and ecoregions, but we also know that common environmental changes experienced by nearby populations can induce synchronous fluctuations in their demographic parameters." Maybe cite Schaub paper at the end?

L62: The Moran effect is a special case of environmental synchronization, so I would rephrase “dispersal and the Moran effect” to “dispersal and environmental effects, including the Moran effect, which proposes …”

L75 (Fig 1. Legend): Please specify: synchrony of what? (Some demographic parameter.)

L82-83: Suggest shortening: "two intraspecific passerine populations (i.e., two populations of the same species)" to "two populations of the same passerine species".

L91: Suggest changing “spatially large scales” to “large spatial scales”.

L102: here you say “few studies” but I thought in the Abstract you say that none had?

L148-150: Can you make it clear that the 7 MBJV ecoregions are smaller units that fall within the broader EPA ecoregion II? And can you avoid using the word “ecoregions” for both the EPA and the MBJV units? Perhaps you can refer to the EPA units as “regions”. Further, I think it may be clearer to list only the 2 MBJV ecoregions that you did include and simply describe why you excluded others without naming them.

L153: The line about EPA Level III and IV should appear immediately after the sentence stating that you restricted the study to EPA Level II. I suggest rewording it as follows: “EPA Level III and IV ecoregions were not included as they are smaller…”

L164: change “considered” to “calculated”

L167: Should this read: “The only nesting parameter not available…” instead? Also, is it really the “only” one? I suggest changing that to a simple statement saying that clutch size was not available.

L169-171: I suggest deleting: “We estimated parameters for AC bluebirds and chickadees, and CH bluebirds and chickadees.” And replacing it with “We estimated each parameter separately for each species and ecoregion.” Or else including that information elsewhere.

Table 1: I don’t think “Range” is the appropriate name for a column that describes “number of nests”. Maybe “n per year” would be more appropriate?

L184-185: “and this led to unrealistic annual parameter estimates (e.g., 100% hatchability every year).” It isn’t clear to me why a longer nest-check interval would lead to estimates of 100% hatchability every year. I suggest you simply explain why you used a simpler method (simple GLM vs. logistic exposure GLM) based on the data limitations you describe (i.e., a long check interval isn’t recommended for that technique, a point which could be reinforced by citing the original Shafer 2004 paper). You could also state in that sentence why you believe the resulting (upward) bias in estimates is not a major concern. That is, because it is likely consistent across regions and species.

L189-190: I suggest deleting the sentence “The model with nest location…”. You can include such a justification in the Table footnotes perhaps? But I don’t think it is necessary.

L196: The first mention of the word “canalized” is unclear without a definition. I suggest replacing the first mention with the actual meaning of the word. Then you can explain that this is part of the environmental canalization hypothesis.

L208: “closer to one” should be “closer to or above one” as (at least CV) can exceed 1 if variability is very high.

L233-235: Minor point, but I think it would be just as easy (and maybe more meaningful) to simply state how many nests of each species occurred in each region (4 numbers total).

L251: typo: “specie” should be “species”

L264-267: “However, we detected increasing trends in positive correlations…” I think this is a most interesting result and provides a lot of “food for thought” for potential causes the Discussion. Can you provide statistical evidence for the trends? Slope estimate and CI or p value? If “significant”, it seems like this is a real result that contradicts statements elsewhere that no real patterns were observed.

L313: I don’t think you can state “potentially a spurious effect” here, parenthetically, without providing evidence or reasoning.

L322: can you remind the reader why it was not expected? Is this because of the canalization hypothesis?

L328: “synchrony is occurring” This language strikes me as unnecessarily binary. The synchrony may be too weak to detect.

L337: “this one negative trend toward increased synchrony” … please explain again to what negative trend you are referring here. It was introduced in the previous paragraph and “this” is ambiguous. Also, how do you reconcile this statement with your discussion of a positive trend in hatchability synchrony above. Can you be more precise or consistent with your language regarding these “trends”?

Figure 3. The labels across the top (i.e., the parameters) are very helpful. Similar labels for the species/ecoregion across the left side are needed to make the figure easy to interpret without frequently referring to the legend.

7. PLOS authors have the option to publish the peer review history of their article (what does this mean?). If published, this will include your full peer review and any attached files.

Reviewer #1: **Yes: **Meelyn Mayank Pandit

Reviewer #2: No

---

## [Author Response · Author response to Decision Letter 1]

24 Oct 2022

Reviewer 2

Thank you for taking the time to review our revised manuscript and providing your helpful comments. We incorporated your suggestions to clarify our language and conducted a careful read-through to remove typos. We also provided the statistical results of our synchronous trends as suggested in comment 24. Finally, we addressed your other suggestions. Please note that line numbers in your comments and our responses refer to those in the document with track changes (Revised Manuscript with Track Changes_final.docx).

1. L24: I suggest deleting the phrase “inter- and intraspecific”. Or else changing to “inter- and intraspecific interactions among wildlife populations”. Otherwise, the meaning is unclear.

a. We agree that the meaning behind this original phrasing is unclear and specified that the referenced implications are for interactions among and within wildlife populations. 

2. L28: typo: double comma

a. Thank you for pointing this out. We have removed the extra comma. 

3. L40: “…study species may be resilient to climate change”: Can you insert somewhere earlier in the abstract that weather/climate is a common driver of spatial synchrony in population dynamics? That would make this statement seem more believable, connecting the dots between climate change and population synchrony.

a. That’s a great suggestion! We added a statement in lines 24–25 noting that variations in climate are a common drive of spatial synchrony. 

4. L49-50: This sentence requires refinement, perhaps as follows: “Studies focusing on sub-populations and smaller regions are common, but are inadequate for understanding population dynamics across a species' range.”

a. Thank you for this suggestion. We feel that this revised sentence is more concise and accurately reflects the intent of the original statement. 

5. L54: typo: double period after “United States declined [5]”

a. We removed the extra period. 

6. L55-58: Most widespread populations exhibit both areas of high and low spatial synchrony within populations (e.g., Allen et al. “Mapping spatial synchrony…”, Conservation Bio, 2021). I suggest revising to something like the following: “...likely vary among populations and ecoregions, but we also know that common environmental changes experienced by nearby populations can induce synchronous fluctuations in their demographic parameters." Maybe cite Schaub paper at the end?

a. This is a great suggestion that clarifies our original statement! We made the suggested revisions and cited Schaub et al. (2015). 

7. L66: The Moran effect is a special case of environmental synchronization, so I would rephrase “dispersal and the Moran effect” to “dispersal and environmental effects, including the Moran effect, which proposes …”

a. We specified that the Moran effect is one type of environmental synchronization. 

8. L78 (Fig 1. Legend): Please specify: synchrony of what? (Some demographic parameter.)

a. Thank you for pointing this out. We specified the caption that the scenario depicts various fluctuations in a given demographic parameter. 

9. L87-88: Suggest shortening: "two intraspecific passerine populations (i.e., two populations of the same species)" to "two populations of the same passerine species".

a. For brevity, we incorporated your suggested wording into this sentence. 

10. L96: Suggest changing “spatially large scales” to “large spatial scales”.

a. We incorporated your suggested wording into this sentence. 

11. L107: here you say “few studies” but I thought in the Abstract you say that none had?

a. We apologize for this discrepancy. When we drafted our original manuscript, we were unaware that other studies had used citizen science data to assess if demographic parameters fluctuated in synchrony. Another reviewer pointed out examples of these studies, and we revised the statement in the Introduction but forgot to do so in the Abstract. We modified the Abstract (line 28) to state that a few studies have studied this phenomenon. 

12. L152-155: Can you make it clear that the 7 MBJV ecoregions are smaller units that fall within the broader EPA ecoregion II? And can you avoid using the word “ecoregions” for both the EPA and the MBJV units? Perhaps you can refer to the EPA units as “regions”. Further, I think it may be clearer to list only the 2 MBJV ecoregions that you did include and simply describe why you excluded others without naming them.

a. MBJV ecoregions are not always smaller than EPA ecoregions, and their boundaries do not always match those of EPA ecoregions. For example, the MBJV’s Atlantic Coast ecoregion encompasses the entirety of Florida and Maine, whereas under the EPA Level II model, these states contain three and two ecoregions, respectively.

b. We agree using the term “ecoregions” for the MBJV and EPA units is confusing. Because “ecoregions” is the proper term for the EPA units, and to avoid confusing readers who are familiar with the EPA model, we replaced the word “ecoregion” with “region” when referring to the MBJV model. We also removed the unused MBJV regions from lines 154–155 and stated in lines 160–162 that we considered an additional five regions but excluded them because of lack of data. 

13. L156: The line about EPA Level III and IV should appear immediately after the sentence stating that you restricted the study to EPA Level II. I suggest rewording it as follows: “EPA Level III and IV ecoregions were not included as they are smaller…”

a. We apologize for the misunderstanding, but we did not use EPA Level II ecoregions in our study. To avoid confusion, we clarified this in the text (line 156) and stated our reasons why we used the MBJV regions instead (lines 156–160).

14. L172: change “considered” to “calculated”

a. We made the necessary correction. 

15. L175: Should this read: “The only nesting parameter not available…” instead? Also, is it really the “only” one? I suggest changing that to a simple statement saying that clutch size was not available.

a. We apologize for the confusion of our original wording. We meant to state that although clutch size data were available, we did not consider this parameter because it typically varies more between the first and last clutch of a season for bluebirds than it does between years. We rephrased the sentence to read “The only nesting parameter that we did not consider was clutch size…”

16. L177-179: I suggest deleting: “We estimated parameters for AC bluebirds and chickadees, and CH bluebirds and chickadees.” And replacing it with “We estimated each parameter separately for each species and ecoregion.” Or else including that information elsewhere.

a. We apologize if our original wording was not clear. We cannot make the suggestion as is because although we estimated nesting parameters for each species, we did not estimate these for each region. We revised the sentence to state that we estimated nesting parameters for each species and each species-region combination. 

17. Table 1: I don’t think “Range” is the appropriate name for a column that describes “number of nests”. Maybe “n per year” would be more appropriate?

a. We apologize for the confusion. The values in the column “Range” reflect the minimum and maximum number of nests used in analyses per year, as the number of observations were not consistent across time. We changed the caption of Table 1 (line 180) to clarify what these values refer to. 

18. L194-195: “and this led to unrealistic annual parameter estimates (e.g., 100% hatchability every year).” It isn’t clear to me why a longer nest-check interval would lead to estimates of 100% hatchability every year. I suggest you simply explain why you used a simpler method (simple GLM vs. logistic exposure GLM) based on the data limitations you describe (i.e., a long check interval isn’t recommended for that technique, a point which could be reinforced by citing the original Shafer 2004 paper). You could also state in that sentence why you believe the resulting (upward) bias in estimates is not a major concern. That is, because it is likely consistent across regions and species.

a. We modified our statement to specify that the logistic-exposure method recommends nest check intervals of 3–5 days, and our nest intervals spanned a larger range (lines 192–194). 

b. Regarding the statement that the upward bias in our estimates is not a concern, we discussed this in lines 197–199.

19. L200-202: I suggest deleting the sentence “The model with nest location…”. You can include such a justification in the Table footnotes perhaps? But I don’t think it is necessary.

a. We removed this sentence from the Methods and added it to the caption of S1 Table 1 (lines 558-560).

20. L207: The first mention of the word “canalized” is unclear without a definition. I suggest replacing the first mention with the actual meaning of the word. Then you can explain that this is part of the environmental canalization hypothesis.

a. Thank you for this suggestion. To make the meaning clearer to readers, we added a short definition of the term (lines 206–207) before referring to the term itself. 

21. L218: “closer to one” should be “closer to or above one” as (at least CV) can exceed 1 if variability is very high.

a. We added a statement noting that values of or greater than one are possible for CV and indicate high variability. 

22. L245-249: Minor point, but I think it would be just as easy (and maybe more meaningful) to simply state how many nests of each species occurred in each region (4 numbers total).

a. Thank you for the suggestion. We added the number of nests found at the species and region levels. 

23. L266: typo: “specie” should be “species”

a. We changed “specie” to “species”. 

24. L286-288: “However, we detected increasing trends in positive correlations…” I think this is a most interesting result and provides a lot of “food for thought” for potential causes the Discussion. Can you provide statistical evidence for the trends? Slope estimate and CI or p value? If “significant”, it seems like this is a real result that contradicts statements elsewhere that no real patterns were observed.

a. This is an excellent suggestion! We included synchrony trend methods and results using linear models of correlation coefficients against time (i.e., sliding windows; lines 241–243 and lines 275–313). 

b. Though some of these trends were significant and showed increasing or decreasing patterns in synchrony, this does not contradict other statements (lines 35, 332, and 354) that we did not observe consistent patterns across parameters, species, and regions.

25. L342: I don’t think you can state “potentially a spurious effect” here, parenthetically, without providing evidence or reasoning.

a. We removed this phrase. 

26. L352: can you remind the reader why it was not expected? Is this because of the canalization hypothesis?

a. This is an excellent suggestion! We explained that we expected to find little synchronization because nesting parameters for short-lived and highly productive passerines are typically canalized against temporal variability, and therefore would be expected to show little synchrony as well (lines 352–353). 

27. L359-360: “synchrony is occurring” This language strikes me as unnecessarily binary. The synchrony may be too weak to detect.

a. The synchrony may indeed be too weak to detect, but because we only found one parameter with a significant overall correlation, we cannot definitively state that synchrony is or isn’t occurring. We added a statement specifying that if synchrony is present, we cannot detect it. 

28. L373: “this one negative trend toward increased synchrony” … please explain again to what negative trend you are referring here. It was introduced in the previous paragraph and “this” is ambiguous. Also, how do you reconcile this statement with your discussion of a positive trend in hatchability synchrony above. Can you be more precise or consistent with your language regarding these “trends”?

a. In light of our new synchrony trend analyses per the suggestion in comment 24, we reworded this part of the Discussion to include our new results (lines 334–338, 369–373).

29. Figure 3. The labels across the top (i.e., the parameters) are very helpful. Similar labels for the species/region across the left side are needed to make the figure easy to interpret without frequently referring to the legend.

a. Thank you for this suggestion. We added labels for each level of species and region to make it easier for readers to view our results.

---

## [Editor Report · Decision Letter 2]

2 Nov 2022

Using citizen science to determine if songbird nesting parameters fluctuate in synchrony

PONE-D-22-01496R2

Dear Dr. Harrod,

We’re pleased to inform you that your manuscript has been judged scientifically suitable for publication and will be formally accepted for publication once it meets all outstanding technical requirements.

Sincerely,

Charles R. Brown

Academic Editor

PLOS ONE
---

## [Editor Report · Acceptance letter]

7 Nov 2022

PONE-D-22-01496R2 

Using citizen science to determine if songbird nesting parameters fluctuate in synchrony 

Dear Dr. Harrod:

I'm pleased to inform you that your manuscript has been deemed suitable for publication in PLOS ONE. Congratulations! Your manuscript is now with our production department. 

Kind regards, 

on behalf of

Dr. Charles R. Brown 

Academic Editor

PLOS ONE